# Analysis of Catecholamines and Pterins in Inborn Errors of Monoamine Neurotransmitter Metabolism—From Past to Future

**DOI:** 10.3390/cells8080867

**Published:** 2019-08-09

**Authors:** Sabine Jung-Klawitter, Oya Kuseyri Hübschmann

**Affiliations:** Department of General Pediatrics, Division of Neuropediatrics and Metabolic Medicine, University Hospital Heidelberg, 69120 Heidelberg, Germany

**Keywords:** inborn errors of metabolism, catecholamines, pterins, HPLC, fluorescence detection, electrochemical detection, MS/MS

## Abstract

Inborn errors of monoamine neurotransmitter biosynthesis and degradation belong to the rare inborn errors of metabolism. They are caused by monogenic variants in the genes encoding the proteins involved in (1) neurotransmitter biosynthesis (like tyrosine hydroxylase (TH) and aromatic amino acid decarboxylase (AADC)), (2) in tetrahydrobiopterin (BH_4_) cofactor biosynthesis (GTP cyclohydrolase 1 (GTPCH), 6-pyruvoyl-tetrahydropterin synthase (PTPS), sepiapterin reductase (SPR)) and recycling (pterin-4a-carbinolamine dehydratase (PCD), dihydropteridine reductase (DHPR)), or (3) in co-chaperones (DNAJC12). Clinically, they present early during childhood with a lack of monoamine neurotransmitters, especially dopamine and its products norepinephrine and epinephrine. Classical symptoms include autonomous dysregulations, hypotonia, movement disorders, and developmental delay. Therapy is predominantly based on supplementation of missing cofactors or neurotransmitter precursors. However, diagnosis is difficult and is predominantly based on quantitative detection of neurotransmitters, cofactors, and precursors in cerebrospinal fluid (CSF), urine, and blood. This review aims at summarizing the diverse analytical tools routinely used for diagnosis to determine quantitatively the amounts of neurotransmitters and cofactors in the different types of samples used to identify patients suffering from these rare diseases.

## 1. Introduction

Neurotransmitters belong to different classes of chemical substances which are involved in synaptic transmission in the central (CNS) or peripheral (PNS) nervous system. Based on their chemical structure, they can be classified into three groups: (1) Amino acid neurotransmitters (glycine, glutamate, γ-aminobutyric acid (GABA), monoamines/biogenic amines (dopamine, serotonin, norepinephrine, epinephrine)), (2) neuropeptides (oxytocin, vasopressin), and (3) atypical neurotransmitters (neurotrophic factors, cytokines, carbon monoxide (CO), nitric oxide (NO), adenosine, adenosine triphosphate (ATP), endogenous cannabinoids, and opioids). Inherited defects in monoamine neurotransmitter biosynthesis or degradation are rare metabolic diseases. In general, primary monoamine neurotransmitter deficiencies can be grouped into five distinct classes: Defects in (1) monoamine biosynthesis, (2) monoamine catabolism, (3) monoamine transport, (4) tetrahydrobiopterin (BH_4_) biosynthesis or recycling, and (5) chaperone associated defects (for a detailed review see [1]). Genes involved are tyrosine hydroxylase (TH), aromatic amino acid decarboxylase (AADC), monoamine oxidase (MAO-A/MAO-B), dopamine ß-hydroxylase (DBH), vesicular monoamine transporter 2 (VMAT2), and dopamine transporter (DAT) in monoamine biosynthesis, transport, and degradation, and GTP cyclohydrolase 1 (GTPCH), 6-pyruvoyl-tetrahydropterin synthase (PTPS), sepiapterin reductase (SR), pterin-4a-carbinolamine dehydratase (PCD), and dihydropteridine reductase (DHPR) in BH_4_ cofactor biosynthesis and regeneration. Recently, mutations in a co-chaperone (DNAJC12) have also been shown to induce a comparable neurological phenotype [2,3,4]. All of these deficiencies present with a broad clinical spectrum, going from moderate late-onset movement disorders to lethal early-onset encephalopathies. As these clinical symptoms often overlap with features also seen in other neurological diseases, this can lead to delayed diagnosis and treatment initiation. Diagnostic workup is usually based on three different sample types: Urine, cerebrospinal fluid (CSF), and blood/dried blood spots (DBS). CSF still represents the gold standard, as diagnosis relies mostly on the quantification of distinct neurotransmitters or pterins, their precursors, or degradation products in the CSF. Treatment is mainly based on supplementation with the missing cofactors or neurotransmitter precursors. The outcome varies, depending on the underlying disorder, the time of diagnosis, and the commencement of treatment. For further details on the clinical symptoms and the treatment options, we refer the readers to [1]. This review will focus on the analytical tools used to establish a proper diagnosis in inborn errors of neurotransmitter biosynthesis and degradation with focus on monoamine and BH_4_ deficiencies. 

## 2. Biosynthesis of Catecholamines, Serotonin, and Pterins

### 2.1. Metabolism and Biosynthesis of Catecholamines

Catecholamines are comprised of a catechol group constituted by a benzene ring with hydroxyl groups at position 3 and 4 and an amine [5]. Physiologically, there exist the three catecholamines dopamine, norepinephrine, and epinephrine. Biochemically, these monoamine neurotransmitters are generated from aromatic amino acids (Figure 1). Therefore, l-phenylalanine is first converted to l-tyrosine by phenylalanine hydroxylase (PAH), which is then converted to l-3,4-dihydroxyphenylalanine (L-DOPA) by TH, which is the rate-limiting enzyme of the pathway [6]. l-DOPA is decarboxylated by AADC to produce dopamine, which is then converted by DBH to norepinephrine. Subsequently, phenyl ethanolamine-*N*-methyltransferase (PNMT) *N*-methylates norepinephrine leading to the formation of epinephrine [7]. Catecholamines are mainly metabolized by MAO A/B and catechol-*O*-methyltransferase (COMT). First, dopamine is deaminated by MAO A/B to 3,4-dihydrophenylacetaldehyde (DOPAL). DOPAL is subsequently oxidized to 3,4 dihydroxyphenylacetic acid (DOPAC) by aldehyde dehydrogenase (AD). Then, DOPAC is *O*-methylated by COMT, generating the homovanillic acid (HVA) that is used for diagnosis in monoamine neurotransmitter deficiencies and excreted in urine [8]. Dopamine can also be converted by COMT to 3-methoxytyramine (3-MT), which is a biomarker for dopamine release [9]. Moreover, COMT can also convert l-Dopa to 3-methoxytyrosine (also known as 3-methyldopa, 3-MD). Norepinephrine and epinephrine are mainly metabolized by MAO A/B to 3,4 dihydroxyphenylglycoaldehyde (DOPEGAL), which is readily reduced by aldehyde reductase (AR) to 3,4-dihydroxyphenylglycol (DHPG). DHPG is converted by COMT to 3-methoxy-4-hydroxyphenylglycol (MHPG), which is the major metabolite of norepinephrine in the human plasma [6]. In the liver, MHPG is oxidized to form vanillylmandelic acid (VMA), which is the principal end product of norepinephrine and epinephrine metabolism. Furthermore, COMT is able to convert epinephrine and norepinephrine to metanephrine (MN) and normetanephrine (NMN), respectively. Moreover, sulfation and glucuronidation also play a role in catecholamine metabolism [10]. As the exact quantification of free catecholamines, metanephrines, and their sulfate conjugates are highly important, sample storage and preparation have to circumvent any process leading to the deconjugation of catecholamine or metanephrine sulfates. Generally, in human plasma, endogenous catechol concentration varies widely and is based on the balance between the entry rate into and the clearance from plasma. Free plasma catecholamines are rapidly cleared and, therefore, exhibit only a short half-life and very low levels in plasma, whereas the sulfate conjugated metabolites have a generally higher level due to slower clearance [6]. 

### 2.2. Biosynthesis and Metabolism of Serotonin

Serotonin, or 5-hydroxytryptamine (5-HT), chemically belongs to the group of indolamines. It is generated from tryptophan by the successive action of tryptophan hydroxylases (TPH; leading to the formation of 5-hydroxytryptophan) and AADC (Figure 2). 5-HT can be further metabolized by MAO A/B to 5-hydroxyindolacetaldehyde (5-HIAL), which is then used by AR to form 5-hydroxyindolacetic acid (5-HIAA), the most important metabolite of serotonin degradation used for diagnosis in CSF analysis. Nevertheless, 5-HIAL can also be oxidized to 5-hydroxytryptophol (5-HTOL), which is a sensitive marker of recent alcohol consumption in human urine. Moreover, serotonin can serve as building block for the synthesis of melatonin (MT; Figure 2). First, serotonin is converted to *N*-acetyl-serotonin (NAS) by serotonin-*N*-acetyltransferase (SNAT). Subsequently, NAS is further metabolized to MT by *N*-acetyl-serotonin-*O*-methyltransferase (ASMT). Furthermore, TRP is also used as a substrate in the kynurenine pathway (not shown). 

### 2.3. Biosynthesis and Metabolism of Pterins

Chemically, pterins are pteridines containing 2-amino-4-oxo structures. They are essential for the biosynthesis of vitamins and cofactors, including BH_4_ and riboflavin. Two groups of pterins exist: (1) Conjugated pterins which have glutamate or *p*-aminobenzoate linked to the pterin moiety, and (2) unconjugated ones with substitutions at the 6- and/or 7- position of the pterin ring by aliphatic side chains. BH_4_ is the major unconjugated pterin in vertebrates. It is an essential cofactor for PAH, TH, tryptophan hydroxylases (TPH1/2), alkylglycerol monooxygenase (AGMO), and nitric oxide synthases (NOS 1-3). De novo biosynthesis of BH_4_ involves three different enzymes—GTPCH, PTPS, and SPR (Figure 3). In the first and rate-limiting step, GTPCH converts GTP in a Zn^2+^-dependent reaction to 7,8-dihydroneopterin triphosphate, which is then metabolized by PTPS to 6-pyruvoyl-tetrahydropterin in a Mg^2+^ and Zn^2+^-dependent reaction. In the last step, SPR reduces 6-pyruvoyl-tetrahydropterin in an NADPH-dependent reaction to BH_4_. Furthermore, there exists a salvage pathway catalyzed by SPR or carbonyl reductase (CR) and dihydrofolate reductase (DHFR), which is important especially in peripheral tissues. Here, SPR/CR catalyzes the conversion of sepiapterin to 7,8-dihydrobiopterin, which is then converted into BH_4_ by DHFR (salvage pathway; [12]). Moreover, alternative pathways for BH_4_ biosynthesis have also been described. Without a functional SPR, aldose reductase (ADR) and CR metabolize 6-pyruvoyltetrahydropterin into 1′-oxo-2′-hydroxyetrahydropterin, which rearranges spontaneously to sepiapterin. Sepiapterin is reduced by CR to 7,8-dihydrobiopterin, which is then reduced to BH_4_ by DHFR [13,14]. Alternatively, BH_4_ can be produced via 1′-hydroxy-2′-oxoproyltetrahydropterin by 3α-hydroxysteroid dehydrogenase type 2 (AKR1C3) and CR [15]. Regeneration of BH_4_ is mediated by the concerted action of PCD and DHPR (Figure 3). During the hydroxylation reaction catalyzed by PAH, TH, or TPH’s, BH_4_ is oxidized to 4a-hydroxytetrahydrobiopterin/pterin-4a-carbinolamine, which is dehydrated to quinoid-dihydrobiopterin (qBH_2_) by PCD. qBH_2_ is the substrate of DHPR and converted back to BH_4_ in a NADH-dependent reaction. Without PCD activity, dehydration of pterin-4a-carbinolamine can also occur non-enzymatically, but at a very low rate, which is insufficient for the maintenance of normal BH_4_ levels. Hence, pterin-4a-carbinolamine is rearranged to dihydroprimapterin in PCD deficiency and can be measured in urine. In the case of low DHPR activity, qBH_2_ rearranges non-enzymatically to 7,8-dihydrobiopterin, which can be reduced to BH_4_ by DHFR. For a detailed description of the regulation of BH_4_ metabolism, we refer the reader to [16].

## 3. Diagnostic Methods

### 3.1. Quantification of Catecholamines and Catecholamine Metabolites

The quantification and analysis of catecholamines and their metabolites in biological samples and specimen is still analytically challenging due to their very low quantities in biological samples and their high tendency to oxidize. Moreover, the clean-up procedures are still elaborate and labor intensive due to the presence of interfering compounds in the samples and the complexity of the matrices analyzed. Table 1 gives an overview on the different types of pre-treatment used in catecholamine analysis. Samples can either be generated by harvesting a fluid of interest or by extracting tissue. It is important to keep in mind that a specific sampling site cannot provide a representative value of catecholamine concentration for the whole body. For example, in plasma there exists a difference in catecholamine concentration depending on the sampling site (arterial versus venous). These differences are due to local release or extraction of the compounds in different parts of the body [8]. Generally, there are three significant variables affecting the stability of a compound during storage. Namely (1) temperature, (2) sample pH, and (3) storage time. For catecholamines, sample integrity is highly dependent on low pH values, especially for long-term storage [17,18]. The standard method for the separation and quantification of catecholamines is high-performance liquid chromatography (HPLC) coupled with fluorescence (FD), chemiluminescence (CL), electrochemical (ECD), or mass spectrometric (MS) detection. Generally, sample preparation is highly dependent on the detection system used. For example, as several compounds in human plasma exhibit absorption and emission maxima which resemble those of catecholamines, a specific isolation method has to be applied to plasma samples if using FD. 

#### 3.1.1. Pre-processing of Biological Fluids

The extraction of catecholamines using aluminium oxide (alumina) is one of the oldest methods described. Alumina forms cyclic complexes with catechol moieties, thereby isolating all compounds containing this moiety—including catecholamines—from complex mixtures. Unfortunately, all substances carrying a catechol moiety are also bound and eluted. Moreover, alumina must first be activated with hydrochloric acid and then be brought to a pH of 8.6. Furthermore, alumina has to be washed extensively after sample binding to remove interfering substances. Desorption of catecholamines is usually done by addition of acids like perchloric acid [19], phosphoric acid, formic acid [20,21], and acetic acid [22]. Taken together, the different steps required for preparation and the instability of catecholamines in contact with alumina leads to low extraction yields for l-Dopa, dopamine, DOPAC, norepinephrine, and epinephrine. Therefore, the most common sample preparation methodology used to date for biological fluids (CSF, plasma, urine, microdialysis) is solid-phase extraction (SPE). SPE is highly selective and results in high extraction yields [23,24]. It is based on the principles of chromatography, but here most of the components are adsorbed to the stationary phase. Catecholamines are eluted after washing with an adequate elution solvent [23]. Several types of cartridges are available, including phenylboronic acid (PCA) cartridges [25,26], alumina cartridges [27], strong or weak cation exchange cartridges [28], hydrophilic-lipophilic balance cartridges (HLB; [24,29]), and C8, C18, or C30 cartridges [24,29,30]. Another method available is liquid-liquid extraction (LLE). The major drawbacks of LLE lie in its low sensitivity and low extraction yields. Moreover, LLE is time consuming and labor-intensive, as it is based on the extraction first into an organic phase and a second extraction step using an acid phase [30]. Other types of pre-treatment include cationic exchange resins, freeze-drying/lyophilization, and protein precipitation (although never used alone). Table 1 summarizes the different types of sample pre-treatments used for recovery and analysis of catecholamines and their metabolites from biological fluids in humans. 

#### 3.1.2. Pre-Processing of Tissue Samples

Quantification of catecholamines and their metabolites from tissue and cell culture samples is less frequent than the usage of fluid samples from patients. Most of the protocols available for the extraction of catecholamines from tissues and cells are based on using brain or adrenal gland samples. This simply reflects the fact that the main target for the analysis of neuropathologies associated with catecholamine metabolism is the brain, whereas the main site of peripheral catecholamine biosynthesis is the adrenal gland. Extraction of catecholamines from tissues and cells needs tissue homogenization or sonication in the presence of a buffer solution. Usually, phosphate or sodium phosphate-based buffers are used [31,32,33]. Alternatively, perchloric acid is used to disrupt the tissue [34]. Additionally, preservatives are added to the homogenate. These include ascorbic acid [35], 1,4-dithiothreitol (DTT; [31]), or sodium bisulfite [34]. All procedures are carried out at low temperatures (4 °C; [36]) or on ice, and the extracts need to be filtered before injection into the chromatographic system [33,35]. 

#### 3.1.3. Chromatographic Columns, Mobile Phases, and Detection Used

Table 2 summarizes columns, mobile phases, and detection methods used for the quantification of catecholamines. The standard columns used for chromatographic separation of catecholamines are either reversed phase C18 or octadecylsilane (ODS) columns. They are characterized by a relatively short run time based on a fast elution of polar compounds. To avoid excessively fast elution and/or poor peak separation, new columns have been invented with lower density. This allows for a quicker access of the analytes to the pores, thereby elongating retention [59]. Moreover, C18 monolithic columns can also be used for analysis. These columns contain no silica particles, but a porous rod of silica. The second type of column often utilized is the less hydrophobic C8-type column. These columns are used to improve peak separation and resolution [29] and to increase the reproducibility of retention times [60]. Other columns applied include base-deactivated silica (BDS) and C30 stationary phases [47], porous graphitic carbon (PGC) columns [41,42], Kromasyl Cyano columns [61], Allure Basix columns [55], pentafluorophenyl propyl columns [10,26,50], and strong cation-exchange (SCX) columns [62,63,64,65]. Another alternative to traditional phase separations is hydrophilic interaction liquid chromatography (HILIC) or aqueous normal phase chromatography. These columns have a water-rich layer over the polar stationary phase, which leads to interactions of the analytes with the hydrophilic environment and the mobile phase via electrostatic and hydrogen bonds. Frequently, these columns are coupled to MS detection [42,43,44] or coulometric detection [51]. Moreover, during the last years, sub-2 µM particle columns have also become more often used for ultra-fast HPLC applications, in combination with amperometric or MS detection—for example, in human plasma and brain tissue [66,67]. 

There are several important parameters which have to be considered when looking for the optimal mobile phase. These include the composition of the buffer solution (qualitatively and quantitatively), the use of an ion-pairing reagent, the use of an organic modifier, and the used pH value [63]. The pH of the mobile phase is dependent on the ionization degree of the analytes, which depends on the pK_a_ values [34]. Depending on the pK_a_ values of the analytes and the pH of the solvent, differentially charged forms of the analytes can be present with different chromatographic behaviors. Therefore, the pH of the mobile phase must be checked before chromatography to avoid that small pH shift inducing large shifts in retention time or peak shape [46]. In the literature, pH values ranging from pH 2.32 to 7.5 have been described for analysis. 

Ion-pairing reagents are usually used to improve the retention time of basic, but not acidic, catecholamines and the chromatographic separation of ionizable compounds in reversed-phase HPLC [22]. Higher concentrations of these reagents increases not only the retention times of basic analytes [34,68], but also the column equilibration time and the costs, due to higher demand of cleaning the system because of high salt content. 

When comparing the different methods described in the literature, isocratic elution is more commonly used than gradient elution, probably due to the fact that this method is simpler and does not depend on extra time for the re-equilibration of the column between two runs. Furthermore, isocratic elution circumvents the baseline drift, which can be found when using gradient elution. Nevertheless, if peak resolution is weak under isocratic conditions, gradient elution can be the solution. 

Detection of catecholamines can be performed using diverse detection methods. These include ECD, FD, CLD, and MS detection [69]. Catecholamines and their metabolites are electroactive compounds, which can be easily detected using ECD. ECD is considered to be highly sensitive and selective [7,61,70]. Classically, the detector includes a working electrode operated versus a reference electrode, and a counter electrode. There are two forms of ECD: Coulometry and amperometry. In both methods the current generated by the reaction is directly proportional to the concentration of the respective analyte in the solution. In coulometry, the majority of the electroactive species is oxidized or reduced. In contrast, in amperometry, this is only the case for a fraction of the electroactive compounds. Therefore, coulometric detection is more sensitive [29] and specific, and has a lower limit of detection, which allows for smaller sample volumes. A coulometric detector consists of one analytical cell with two electrodes at different voltages, which can be coupled to a guard cell [20,22,37] or a conditioning cell [24,46]. As catecholamines reversibly oxidize, a two-electrode detector enhances sensitivity by oxidizing the compounds at the first electrode and then reducing the compounds back at the second electrode. This coupling can also be used to get rid of substances without a reversible oxidation process, like ascorbate and oxygen [29]. Moreover, ECD does not need any derivatization in advance and therefore, the pre-treatment is often simpler when a HPLC step is combined with ECD [44]. However, electrochemical detectors are highly susceptible to fluctuations in the pumping rate (increase in signal-to-noise ratio) and can be impaired by electrode fouling, which is induced when functional groups of an analyte cause its adsorption to the electrode [61]. As this can lead to an attenuation of the signal, it is necessary to perform electrochemical cleaning of the electrode to avoid a drastic reduction in overall sensitivity [71]. 

Catecholamines emit a native fluorescence in UV light, but this is not sufficient for quantification in real biological samples [7,72]. Therefore, as catecholamine concentration and emission intensity are low, usually a derivatization step (pre- or post-column) is included when fluorescence detection is used. Again, the derivatization procedure has to be optimized concerning the concentration of the derivatization agent used, the temperature, and the time of reaction [31,72]. Different derivatization agents are described in the literature. They can be specific for certain functional groups, such as the carboxylic acid group or the amino group (2-(perfluorooctyl)-ethylisocyanate), or recognize specific chemical structures, like benzylamine for 5-hydroxyindols [73]. Classically, the catechol group is first oxidized and subsequently either tautomerized to form trihydroxyindol (THI) derivatives or used for the reaction with *meso*-1,2-diphenylethylenediamine (DPE) or ethylenediamine. THI cannot be used for derivatization with dopamine [54,74], but is well suited for the detection of epinephrine and norepinephrine. In contrast, DPE is highly sensitive and selective [7,75] and has been used for the quantification of catecholamines in plasma [76], urine [77], brain tissue, and dialysate [68,72], amongst others. Unfortunately, reaction products are only stable for 30 min [74], but addition of glycine to the reaction leads to an acceleration and the solution is at least stable for two weeks at room temperature [75]. Ethylenediamine can also be used for post-column derivatization combined with electrochemical oxidation [31,32,78,79]. Moreover, other derivatization reagents have also been described in literature, including benzylamine [80], fluorenylmethyloxycarbonyl chloride (FMOC-CI; [74]), and terbium (III) chloride [54]. Depending on the derivatization reagent used and the derivative generated, excitation and emission wavelengths differ. Usually, ethylenediamine derivatives have an excitation wavelength of 430 nm and an emission wavelength of 505 nm [79,81]. DPE derivatives can be excited at a wavelength of 345 nm and emit at 480 nm [35,75]. In contrast, FMOC-CI derivatives are characterized by excitation and emission wavelengths of 200 nm and 300 nm, respectively [74]. Taken together, fluorescence detection is sensitive and selective and considered to be more reliable than ECD, because the analytical system is less prone to interference and is simpler [35,75], but due to the derivatization step more time-consuming and laborious. 

Chemiluminescence detection with HPLC is also highly selective and sensitive, but needs a post-column system that can initiate and detect the chemoluminescence [53]. The most commonly used detection method is peroxyoxalate chemoluminescence (POCL), which is based on the oxidation of an aryl oxalate ester together with a fluorophore. After HPLC, the catecholamines are electrochemically oxidized, thereby generating *o*-quinones. These then react with ethylenediamine, and these fluorophores are combined with the chemiluminescence solution and detected [82]. In some publications, inhibition of chemiluminescence of oxidized luminol by catecholamines has also been used for detection [49]—a technique which is only suitable for urine samples, but not plasma [60]. 

MS detection associates the retention times of specific compounds with structural information based on the mass-dependent transition between precursor ion and product ions in tandem mass spectrometry (MS/MS). This allows for the quantification and specific identification of the compounds. Usually, acidic analytes like VMA, DOPAC, DHMA, HVA, and MHPG are detected in the negative ion-electrospray mode, whereas basic analytes, including l-Dopa, dopamine, 3-MT, MN, NMN, norepinephrine, and epinephrine, are detected in the positive mode [36,61]. Sensitivity depends on efficient ionization, which can be affected by different variables, including chromatographic variables, the characteristics of the analyte itself, and matrix interference (matrix effect; [55]). It has to be kept in mind that very polar and small molecules are susceptible to ion suppression, which means that at the ionization source there is a competition for ionization efficiency between the analytes of interest and other species in the sample, which can lead to reduction in precision, accuracy of measurement, and limit of detection if both elute at a similar time point. It has been shown that molecules with a higher mass can suppress the signal of smaller molecules. Therefore, suppression or enhancement of ionization by a matrix effect and influence of the mobile phase on ionization of catecholamines have to be monitored when establishing a new method. Nevertheless, MS and MS/MS detection methods are versatile alternatives to ECD, FD, or CLD as they are capable of unequivocal identification of specific analytes [83] and can be automated.

#### 3.1.4. Analysis of Catecholamines in CSF

To ascertain the diagnosis of an inborn error of biogenic amine metabolism, the first investigation is the analysis of CSF profiles for HVA, 5-HIAA, and 3-OMD. As there is a rostrocaudal gradient for HVA and 5-HIAA, it is essential for the sampling of CSF that the first 0.5 mL of fluid from the spinal tap is collected for analysis, immediately frozen at bedside in liquid nitrogen or dry ice, and stored at −80 °C until further analysis. Blood contaminations have to be removed prior to freezing by short centrifugation at 4 °C, as hemolysis will result in the degradation of HVA and 5-HIAA. All metabolites are stable for up to five years in frozen CSF samples at −70 °C, without antioxidants [87] and without protection from light [88]. Generally, HVA and 5-HIAA can be used as indirect markers to explore the functionality of the dopamine and serotonin pathway in the brain (Table 3). Moreover, the analysis of 3-OMD and MHPG (as marker metabolites for dopamine and norepinephrine pathways), together with 5-HTP (as a serotonin precursor), have been proven to be quite valuable for the identification and differentiation of the underlying biogenic amine disorder in one single analysis. Classically, HVA, 5-HIAA, and 3-OMD are analyzed via reverse-phase high HPLC on an ODS column, followed by electrochemical detection. The analytes elute on a reverse-phase HPLC column in order of increasing hydrophobicity. The mobile phase at an acidic pH suppresses ionization of the analytes. Molecules with a catechol ring remain positively charged, even at pH 2.0. Therefore, an ion-pairing reagent (sodium octane sulphonic acid (OSS)) is added to the mobile phase to optimize the interaction with the stationary phase. The mobile phase usually contains methanol to decrease the retention time of the analysis. As CSF does not contain a lot of compounds disturbing the analysis, only a little sample preparation is required before analysis. Nevertheless, as these compounds are very easy to oxidize, a careful sample collection and storage is absolutely required. In 2017, [39] published a method for the determination of HVA, 5-HIAA, 3-OMD, and MHPG in CSF without the need for time-consuming pre-processing of the samples. Here, CSF freshly frozen at bedside without blood contaminations is directly diluted in the mobile phase, shortly centrifuged, and filtered through a nylon filter to remove greater particles and contaminants. Then, the sample is directly injected into the HPLC and analyzed using ED. Typical CSF profiles in inborn errors of biogenic amine metabolism are summarized in Table 3.

#### 3.1.5. Analysis of Catecholamines in Urine

For analysis in urine, the samples should have a pH of less than 4.0 [17,18], as this type of acidification is an effective means to prevent the decomposition of catecholamines. It should be performed after voiding or within 24 h after voiding [89]. Usually, sulphuric acid, acetic acid, or hydrochloric acid are used. Generally, acidification is the best method for prolonged preservation [90]. Nevertheless, acidification below pH 2.0 has to be avoided as this would lead to the hydrolysis of conjugated forms of catecholamines, thereby increasing the concentration of the free forms [18]. Alternatively, antioxidants can be added during sample collection and the acidification step can be done later [89]. Some publications report that addition of EDTA or sodium metabisulfite is also sufficient to stabilize catecholamines in urine samples for two weeks at −20 °C [91]. As urinary levels of HVA and 5-HIAA can vary considerably, not only due to an underlying genetic defect, but also caused by external factors such as neuroblastoma, pheochromocytoma, or other neural crest tumors showing high HVA besides high vanillylmandelic acid and carcinoid tumors producing serotonin and 5-HIAA, these metabolites are usually not used for the diagnosis of a genetic defect. 

#### 3.1.6. Analysis of Catecholamines in Blood Samples

Several protocols for the measurement of catecholamines in blood have been published (Table 2). Generally, blood samples (2 mL) should be centrifuged at 4 °C immediately or at least within 1 h after collection for the separation of plasma from the blood cells [46,47,49,82,92]. The plasma should be transferred to a new vial and immediately frozen at −20 °C. In the past, different antioxidants have been added to the sample prior to centrifugation, but today, most laboratories do not include an antioxidant. Without addition of an antioxidant, catecholamines are stable at −20 °C for six weeks, or for one year at −80 °C. Therefore, plasma should always be stored at −80 °C if possible [17]. Ideally, blood samples (2 mL) should be collected into a tube containing an anticoagulant (EDTA or heparin), transported on ice, centrifuged at 4 °C, and plasma stored at −80 °C until measurement. 

#### 3.1.7. Enzyme Activity Assays

TH activity has been determined in tissue samples [93] and on overexpressed protein extracts generated in *Escherichia coli* [94,95]. Classically, the enzymatic activity is determined using radioactively labelled l-tyrosine [93,96,97]. In the meantime, non-radioactive assays have also been developed measuring the formation of l-tyrosine by TH, either based on HPLC coupled to fluorimetric detection [95] or plate-reader based assays [94]. The first assay to determine AADC activity was published in 1985 by [98]. Here, the activity was determined based on the formation of dopamine, which was detected via HPLC and ED. 

### 3.2. Quantification of Pterins

Measurement of pterins in CSF, blood, and urine is the most common method used to screen for inborn errors of BH_4_ metabolism. Moreover, neopterin can serve as a marker for immune activation. BH_4_ deficiencies in humans are the result of defects in the metabolism or recycling of BH_4_ as a consequence of mutations in the abovementioned enzymes involved in BH_4_ metabolism. The resulting lack of BH_4_ is associated with neurological aggravation, including central hypotonia, progressive mental and physical retardation, peripheral spasticity, seizures, and, in some cases, microcephaly [99]. Diagnosis is based on different biochemical and analytical approaches, which depend on the mode of inheritance and the underlying enzyme defect. All newborns presenting with plasma levels of phenylalanine higher than 120 µmol/L and children with suspicious neurologic symptoms should be analyzed for the presence of a BH_4_ deficiency. Usually, these tests include the measurement of pterins in urine and/or DBS, the measurement of DHPR activity in DBS, a BH_4_ loading test, measurement of pterins and biogenic amins in CSF, and the determination of enzyme activity [100,101,102]. The first two tests are essential and sufficient to differentiate all BH_4_ defects presenting with hyperphenylalaninemia. 

Diagnostically relevant pterins include BH_4_, dihydrobiopterin (BH_2_), biopterin, and dihydroneopterin—but also neopterin, sepiapterin, and primapterin. Each BH_4_ defect has its unique pterin pattern (Table 6 based on personal communication with the “BH_4_ deficiencies guideline” group of the “International Working Group on Neurotransmitter Related Disorders” (iNTD; www.intd-online.org)). In patients suffering from autosomal-recessive GTPCH deficiency, the amounts of neopterin and biopterin are decreased, whereas in autosomal-dominant forms, normal concentrations can be measured. PTPS deficiency is characterized by very high levels of neopterin and very low amounts of biopterin. In contrast, sepiapterin deficiency shows no changes in neopterin and biopterin levels, but high levels of BH_2_. Moreover, [103] could show that patients suffering from SRD also show high levels of sepiaterin not only in CSF, but also in urine, thereby providing a new non-invasive diagnostic tool. DHPR deficiency presents with mostly normal levels of neopterin and biopterin. The measurement of DHPR activity should be performed in the case of hyperphenylalaninemia and normal pterin profile. Neopterin measurements are also used for monitoring of HIV therapy, in patients with autoimmune diseases, and in cancer patients. 

Generally, pterins have poor solubility in organic solvents and water. Depending on the substituents and their positions on the pteridine ring, solubility can dramatically change. Pterins are also very easy to oxidize and to reduce. Fully oxidized pterins show either yellow (sepiapterin, 3-hydroxysepiapterin) or blue (neopterin, monapterin, biopterin, pimapterin, pterin, isoxanthopterin) fluorescence, which is used for their measurement in biological fluids [104,105]. Reduced pterins like BH_4_ are extremely sensitive to oxygen and light and dissociate rapidly to pterin, pterin-6-carboxylic acid, isoxanthopterin, and orxanthopterin. In body fluids, most of the naturally occurring pterins are present in their reduced forms and must be oxidized to the highly fluorescent oxidized forms prior to HPLC and further analysis. Classically, pterins are detected using HPLC coupled either to a FL or an EC detector, GC-MS, or MS/MS. The most widely used method for the determination of pterins, especially in CSF, is HPLC coupled to fluorescence detection. This method is quite widely used due to its appropriate sensitivity. 

#### 3.2.1. Pre-Processing of Biological Fluids

For the preprocessing of biological samples, several protocols have been developed (Table 4). As mentioned above, BH_4_ and BH_2_ coexist in biological samples but are hardly fluorescent. Therefore, their direct determination is not possible. To prevent BH_4_ and BH_2_ from oxidizing before analysis, different antioxidants like dithioerythritol (DTE), DTT, or diethylenetraminepentaacetic acid (DETAPAC) are usually included into the samples [106,107,108]. For analysis, BH_4_ and BH_2_ are indirectly determined as strongly fluorescent biopterin and pterin after pre-column oxidation using either iodine [109,110] or MnO_2_ [111]. Iodine oxidation consists of two independent steps. By acidic iodine oxidation, BH_4_ and BH_2_ are both oxidized to biopterin, whereas dihydroneopterin is oxidized to neopterin. By using alkaline iodine oxidation, BH_4_ is oxidized to pterin and BH_2_ to biopterin. To determine the amount of BH_4_ in the analyzed sample, the difference between the amounts of biopterin received by acidic and alkaline oxidation is calculated. Therefore, the iodine oxidation method is classically used for differential oxidation of the pterins and the quantification of BH_4_ [109,110]. Oxidation in the presence of MnO_2_ in acidic conditions is a routine method to determine total pterin levels (fully oxidized neopterin, pterin, biopterin, isoxanthopterin, monapterin, primapterin), but does not allow for distinguishing between BH_4_ and BH_2_ [111]. Ref. [112] reported on the generation of biopterin from BH_4_ and BH_2_ by using UV photo irradiation. This method included one off-line irradiation step before injection into the HPLC system. In 2009, the group presented an on-line UV photo irradiation HPLC system using a diode array coupled to a fast scanning fluorescence detector, which directly allowed for the determination of BH_4_, BH_2_, biopterin, pterin, neopterin, and isoxanthopterin simultaneously from the same sample [113]. Generally, most protocols include a precipitation or filtration step before HPLC analysis in order to filter out proteins in the samples which may adhere to the HPLC columns and therefore hinder separation. 

#### 3.2.2. Pre-Processing of Tissue Samples

Pterins are predominantly measured in biological fluids, but there are also protocols available for the determination of pterin levels in cells. Classically, patient-derived fibroblasts or mononuclear cells from patient blood are used to verify diagnosis by performing enzyme specific activity assays (see also below) or measurement of neopterin and biopterin levels directly in the cells. Therefore, the cells are first stimulated with cytokines (classically interferon γ and TNF α or phytohemagglutinin) to induce BH_4_ biosynthesis, and then lysed in a lysis buffer [114,115,116]. To determine neopterin and biopterin in patient-derived fibroblasts, the cell lysate is oxidized under acidic conditions using the iodine oxidation method [109,110], and the generated neopterin trisphophate is dephosphorylated. After a deproteinization step using Ultrafree MC filters, the resulting solution is subjected to HPLC analysis [115]. In 2012, ref. [117] described another preparation method for HUVEC cells. Here, the cells were trypsinized from the cell culture plate and lysed in ice-cold TCA (0.2M) containing 1 mM EDTA, 50 mM ascorbic acid, and 6.5 mM DTE by freezing and ultrasonication. For analysis, they included internal standards into the lysate and applied the samples to HPLC analysis for biopterin. 

#### 3.2.3. Chromatographic Columns, Mobile Phases, and Detection Used

The classical columns used for separation of pterins are either reversed phase C18 or ODS columns (Table 5). Other columns which have also been used for separation include phenyl-hexyl columns [126], AquitiyUPLC HSS T3 columns [124], Partisil-10 SCX columns [130], and Zorbax-Eclipse XDB columns [122]. Also, hydrophilic interaction liquid chromatography (HILIC) can be used, coupled to either MS [119,125] or fluorescence detection [127]. 

In the literature, a long list of different mobile phases has been described (Table 5). Generally, mobile phases containing acetonitrile or methanol—often in combination with low amounts of formic acid and aqueous buffers, like citrate buffer and phosphate buffers—are used for separation. pH values ranging from pH 2.8 to 5.0 have been described for the quantification of pterins. Some protocols also use ion-pairing reagents [117], but most do not include these compounds. Based on the literature, isocratic and gradient elution are equally used for analysis. 

Detection of pterins can be performed with diverse detection methods. In the early days of analysis radioenzymatic assays [135], electrophoresis [136], thin layer chromatography [137], and even a growth factor assay in Crithidia fasciculata [138] were used for determination. Today, most protocols are based on ECD, FD, ECD coupled with FD, and mass spectrometry detection. Especially for the detection of BH_4_ in CSF, together with other pterins, ECD coupled with FD is used [139,140,141]. In that case, BH_4_ is detected electrochemically, whereas BH_2_ and dihydroneopterin are measured via fluorescence after post-column coulometric oxidation into biopterin and neopterin. As CSF also contains a lot of other electroactive compounds, this method for the quantification of BH_4_ is quite challenging [142]. The most commonly used method for the detection of pterins in the diagnostic setup in urine, blood, and DBS is the indirect oxidation method originally described by [109,110], which is based on the high fluorescence of fully oxidized pteridines. As described in Section 3.2.1, the samples have to be oxidized before separation, using iodine/iodide solution under acidic and alcaline conditions. Separation of the different samples is then performed using either an HPLC system with column switching [132], without the need for sample pre-treatments for the analysis of total pterins in amniotic fluid, serum, urine, and CSF, or with a simple isocratic HPLC system coupled to FD. In 2017, ref. [39] published a protocol, allowing for the determination of BH_4_ and other pterins (besides catecholamines and derivates) in CSF without the need for complex pre-preparation steps using HPLC coupled to coulometric and fluorescence detection. Mass spectrometry is also applied to quantify pterin levels in the samples, as it allows for the specific identification of the compounds. Most protocols use the positive ion-electrospray mode for detection [108,119,124,126], but usage of the negative mode has also been described [105,125].

#### 3.2.4. Analysis of Pterins in CSF

As mentioned above, BH_4_ in the CSF is highly sensitive to auto-oxidation. Therefore, correct sample collection and storage is a prerequisite for a valid measurement. Interestingly, it has been suggested that there exists no rostrocaudal gradient for neopterin and biopterin in contrast to the situation described for catecholamines [38]. To prevent auto-oxidation, antioxidants like DTE and DETAPAC should be added to the freshly gathered sample, and the samples should be immediately frozen at −80 °C or dry ice, as was described for catecholamines. For further analysis, large proteins are removed by filtration using different types of filter units, like the Ultrafree 10000 filter unit. Smaller proteins are usually precipitated using TCA precipitation and the samples are oxidized using either the MnO_2_ or iodine/iodide method. This oxidation step is not recommended for the measurement of sepiapterin. Reduced pterins like BH_4_ are often analyzed directly by HPLC coupled to electrochemical detection. Generally, a careful sample collection and storage is absolutely required. Recently, ref. [39] described a method for the determination of pterins in the CSF without the need of time-consuming sample pre-preparation. For measurement, they used HPLC separation coupled to coulometric and fluorescence detection. In brief, freshly frozen CSF samples (200 µL) are prepared without pre-processing. In the case of high protein content or colored samples, the authors advise to filter the samples through 10 kDA centrifugal filter prior to analysis. Subsequently, the sample is injected into the HPLC device coupled to in-line ED and FD. Neopterin, BH_2_, and BH_4_ are separated in the HPLC column. BH_4_ is oxidized by the first electrode in the analytical cell to quinonoid dihydrobiopterin, and reduced to BH_2_ by the second electrode. BH_2_ and total neopterin (neopterin and dihydroneopterin) are measured in the same run by FD. A postcolumn oxidation step oxidizes dihydroneopterin to neopterin, and BH_2_ to biopterin—all of which are analyzed by FD. Table 6 summarizes characteristic pterin profiles measured in CSF depending on the underlying defect.

#### 3.2.5. Analysis of Pterins in Urine

Urinary pterin determination is used for diagnosis in several medical fields, including immunology, oncology, psychobiology, and metabolic disorders. Regarding sample preparation, native urine should be protected from light and stored at least at −20 °C (up to −80 °C) until processed for analysis. Classically, two different procedures for oxidation of urine are commonly used. The first one is the oxidation with MnO_2_ under acidic conditions. This oxidation method is routinely used for the determination of total pterins (fully oxidized neopterin, pterin, biopterin, primapterin, monapterin, and isoxanthopterin). It cannot be used for the quantification of BH_4_. To quantify BH_4_, differential oxidation with iodine/potassium iodide under basic conditions and acidic is necessary [109,110]. The concentration of BH_4_ present in the sample is calculated by determining the difference in biopterin content between the two different oxidation procedures. Most protocols are based on HPLC coupled to fluorescence detection, but there are also newer protocols available combining the separation via HPLC with MS/MS [126,128]. Analysis of urinary sepiapterin content has been shown to be a useful non-invasive tool to identify patients suffering from SRD [103]. Here, morning urine is sampled and an aliquot is used for analysis. To the sample, ascorbic acid is added as antioxidant and then the sample is ultra-filtered prior to injection into the HPLC column using a gradient protocol for separation and subsequent FD. Table 6 gives an overview on characteristic pterin profiles measured in urine depending on the underlying defect.

#### 3.2.6. Analysis of Pterins in Blood Samples

Detection of pterins can also be performed in plasma and serum, but also from dried blood spots (Guthrie cards). Plasma and serum samples should be immediately frozen in dry ice and stored at −80 °C until further analysis. DBS should be dried completely before shipment. If Guthrie cards are used, for every measurement, 4–6 blood spots with a diameter of 6 mm are cut out and the pterins are extracted from the spots. The extract is often ultra-filtered before injection into the HPLC. Classically, no prior oxidation is performed when dried blood spots are analyzed, and analysis is predominantly based on HPLC coupled to fluorescence detection [123]. Recently, a UPLC-MS/MS-based method has also been described by [124]. Analysis of pterin levels in human plasma and serum usually involves oxidation with either MnO_2_ or differential oxidation with iodide/iodine after a filtration step [121]. Recently, a new protocol containing a derivatization step for the precise measurement of BH_4_ has been published by [119]. Here, BH_4_ is directly derivatized with benzoyl-chloride and analyzed by MS/MS. In Table 6 on overview is given on the characteristic pterin profiles measured in DBS depending on the underlying defect.

#### 3.2.7. Enzyme Activity Assays

For the enzymes involved in BH_4_ synthesis and regeneration, enzyme activity essays have been established. GTPCH, the first and rate-limiting enzyme, is expressed in peripheral tissues and in the brain. Its expression can be induced in T lymphocytes, macrophages, or in fibroblasts by cytokines. [143] showed that GTPCH is also measurable in liver tissue. Reference values and age-dependent changes in GTPCH activity in stimulated mononuclear blood cells measured by HPLC have been reported by [116]. Of note, this test must be performed within 20 h after sampling and is quite complex. Generally, interferon-γ has been described to be a valid stimulus for the induction of GTPCH activity in fibroblasts, macrophages, but also THP-1 and T 24 cells [114,115,144]. Tissue samples should be shock-frozen in liquid nitrogen and stored at −80 °C for long term. Tissue lysates should be prepared by grinding the frozen piece of tissue to powder under liquid nitrogen before adding ice-cold homogenization buffer. Repeated freeze-thaw cycles of lysates must be avoided as this will decrease GTPCH enzyme activity. Therefore, activity assays should be performed from fresh lysates. Classically, GTPCH activity is determined by measuring the formation of neopterin, which is generated by complete oxidation and dephosphorylation of dihydroneopterinphosphate, starting from GTP. Conversion of dihydroneopterinphosphate to neopterin is performed by acidic oxidation using iodine/iodide, followed by dephosphorylation using alkaline phosphatase (at pH 8.5–9.0), and generated neopterin is detected fluorometrically at 350/440 nm after HPLC separation [145,146]. Reference values for cytokine-stimulated fibroblasts have been established [147], and are 2.6 µU/mg protein for control cells, 0.8 µU/mg protein for autosomal-recessive GTPCH deficiency, and 0.4 µU/mg protein for l-Dopa responsive dystonia (DRD). Non-stimulated fibroblasts have no detectable GTPCH activity. 

PTPS, SR, and DHPR are constitutively expressed in different cell types, including fibroblasts. Activity assays for all three enzymes in erythrocytes, amniocytes, fibroblasts, and lysates of tissue samples have been published [115,148,149,150,151]. Generally, to measure PTPS activity in erythrocytes, heparinized blood should be used. If only EDTA-treated blood samples are available, manganese must be added in excess to ensure full activity of the manganese-dependent PTPS. Furthermore, to avoid oxidation of the generated BH_4_ in the assay, O_2_ has to be displaced from hemoglobin (Hb) using CO. In the assay, PTPS converts 7,8-dihydroneopterintriphosphate to 6-pyruvoyltetrahydropterin, which is further metabolized by SPR in the reaction buffer to BH_4_. The BH_4_ is then chemically oxidized to biopterin, which is quantified using HPLC coupled to FD or MS/MS. Reference values for unstimulated dermal fibroblasts are 0.7 µU/mg protein for healthy controls and <0.05 mU/mg protein for autosomal recessive PTPS. For erythrocytes, normal values for PTPS are 35–77 µU/g Hb (fetus), 34–64 µU/g Hb (newborns up to one month), and 11–29 µU/g Hb (children and adults), respectively. In amniocytes, the reference value is 3.0 µU/mg protein [147]. 

To measure the activity of SPR, conversion of sepiapterin to BH_2_ is used, followed by an oxidation step in the presence of iodine solution, leading to the formation of biopterin which can easily be detected fluorometrically [148]. Reference values for SPR activity are 138 µU/mg protein for non-stimulated dermal fibroblasts, <10 µU/mg protein for autosomal recessive SPR deficiency, and 143 µU/mg protein for amniocytes, respectively [147]. 

DHPR activity has been determined in peripheral blood cells, erythrocytes, fibroblasts, amniocytes, and tissue lysates [115,149,152,153]. Activity of DHPR is measured by monitoring the reduction of NADH during catalysis at 340 nm. This assay is highly sensitive but, due to interference with Hb, not recommended for the analysis of erythrocytes. Here, the non-enzymatic coupling of the oxidation of 6-methyltetrahydropterin to quinonoid 6-methyldihydropterin with the reduction of ferricytochrome c to ferrocytochrome c is recommended and measured at 550 nm [149]. Reference values for DHPR activity in unstimulated dermal fibroblasts are 6.7 mU/mg protein in control cells, <0.3 mU/mg protein in autosomal recessive DHPR deficiency, 7.5 mU/mg in amniocytes, and 1.8–3.8 mU/mg Hb in dried blood spots with a decrease in activity observed after one year of life (from 5.4–8.9 mU/mg to 4.2–7.0 mU/mg; [147]). Clinically, the measurement of DHPR activity in dried blood spots is more relevant, since pterins in urine or in blood can be normal in DHPR deficient patients [149,154].

PCD activity has first been analyzed in rat liver homogenates [155]. For analysis, 6(S)-propyl-4a-hydroxy-tetrahydropterin is used as substrate in the presence of excess DHPR and NADH and the reduction of NADH during catalysis is monitored at 340 nm [155,156]. The assay has also been used for the characterization of patient-specific mutations in PCD in an *E. coli*-based overexpression system [157], and to determine the activity of PCD in human fetal tissues [158] and duodenal mucosa [159]. Currently, no good reference values are available for PCD. 

## 4. Pitfalls

### 4.1. Pre-Analytical and Methodological Pitfalls

Several pitfalls have to be kept in mind when performing the analysis of catecholamines and pterins in biological samples. Sample collection should be performed with caution and attention. Catecholamines are easy to oxidize, and are temperature- and pH-sensitive. Therefore, acidification below pH 2.0 has to be avoided. Pterins are sensitive to UV light and also very easy to oxidize and reduce. Especially the reduced pterin like BH_4_ decompose rapidly to pterin, pterin-6-carboxylic acid, isoxanthopterin, or xanthopterin. Therefore, to prevent BH_4_ and BH_2_ from oxidation before analysis, antioxidants (DTE, DETAPAC) need to be added and the samples should be protected from light. No single sampling side can provide the “real” value of the concentration of the respective metabolite under investigation. For example, in the CSF there exists a rostrocaudal gradient of HVA and 5-HIAA, and in plasma there are significant differences in catecholamine concentrations depending on the site of sampling (arterial versus venous). Furthermore, other endogenous—but also exogenous—compounds like drugs and their metabolites have similar absorption and emission maxima as catecholamines and pterins (for example vitamin B6 and BH_2_).

Critical factors for the acquisition of CSF samples are the time point of sampling (best in the morning), the rostrocaudal gradient (higher concentrations of some metabolites in the final fraction than in the first one), the light sensitivity of some analytes, and the avoidance of blood contaminations. Therefore, a standardized lumbar puncture protocol under the highest clinical standards should be used, and analysis should be performed in a standardized CSF fraction. As CSF leaves the spine dropwise, counting the number of drops might be useful for estimating the sample volume. Samples must be frozen immediately at bedside and stored at −80 °C until analysis to prevent degradation of metabolites. Contaminations with blood have to be removed immediately by centrifugation at 4 °C prior to freezing, as hemolysis will result in a reduced level of HVA and 5-HIAA due to oxidation. In the CSF, pterins, 5-MTHF, and amino acids should always be included into analysis and, depending on the neurotransmitter pattern, also sepiapterin [1]. Moreover, it is important for interpretation to compare the patient’s values to age-matched reference values and to the patient´s own values [39]. Technically, as CSF does not contain as much interfering substances as are present in blood, plasma, urine, or tissue extracts, no extensive purification is necessary. Nevertheless, to ensure a good analytical performance, guard columns and graphite filters should be replaced regularly. Urine samples should always be protected from light until analysis, which should be performed best immediately within 24 h after voiding. To prevent decomposition of catecholamines, urine samples should have a pH of less than 4.0 but more than 2.0. If storage is necessary, urine should be kept at 20 °C up to −80 °C. Dried blood spots should always be dried completely before shipment or analysis. 

For analysis, it is important to keep in mind that the separation conditions used can dramatically change the elution time of a compound. The most important factors include the pH value of the mobile phase and the ion-pairing reagent used. The pH of the mobile phase is related to the ionization degree of the analytes, depending on their pK_a_ values [34]. As a function of the pH of the solvent and the pK_a_ values of the analyte, differentially charged forms of the analytes can be present, which exhibit different chromatographic behaviors. Therefore, it is of highest importance to check the pH of the mobile phase before use in order to avoid small pH shifts, which can induce large shifts in retention time or peak shape [46]. The composition and the pH of the mobile phase is also important for the sensitivity of the electrodes in ECD. When using ECD, it is very important to identify the appropriate setting to achieve complete oxidation of the analyte of interest. The optimal potential is usually identified using a hydrodynamic voltagramm, which relates the generated current with applied potentials [34,47,160,161] after injection of a compound into the HPLC system and subsequent analysis over a range of different voltages with fixed increments [37]. For analysis, the lowest current producing the highest response of the analyte to the electrode should be used as this also reduces the background signal. As pH of the mobile phase can influence these conditions and the redox behavior of the analytes, the optimal pH has to be identified for analysis. The ideal pH value is the result of a balance between the pH value necessary for efficient oxidation of the analytes and the pH value suitable for efficient electrochemical reduction. 

### 4.2. Diagnostic Pitfalls

Correct interpretation of the measurement results discussed above requires detailed clinical information focusing on patient history, prominent clinical symptoms, and radiological and genetic findings, as well as current drug therapy. As an example for measurement results we have included in the appendix example chromatograms for TH- and PTPS-deficiency before and after initiation of the therapy (Figure A1 and Figure A2). There are multiple neurological conditions other than primary neurotransmitter disorders showing alterations in biogenic amines (HVA, 5-HIAA) and pterins (especially neopterin) in CSF. 

Ref. [162] showed in a cohort of 56 infants that there exist patients without a primary defect in neurotransmitter metabolism, but with low levels of HVA and 5-HIAA. Here, HVA deficiency was associated with severe motor impairment and importantly neurodegenerative disorders. In contrast, 5-HIAA levels were decreased in patients with brain cortical atrophy [162]. In line with this, [163] could show in 2013 in their cohort of 1388 patients with neurological disorders that 20% of the cohort showed altered levels of HVA. Low levels of HVA were frequently found in patients with CNS infections, perinatal hypoxic ischemic encephalopathy, mitochondrial disorders, and pontocerebellar hypoplasia type 2. In contrast, high HVA levels were present predominantly in patients suffering from mitochondrial diseases and intraventricular hemorrhage in preterm infants. Moreover, reduced levels of HVA have also been reported in epilepsy [164,165], febrile convulsion [166], postural tremor with dystonia [167], supranuclear palsy [167], dystonia [168](Tabaddor et al., 1978), rapid-onset dystonia-parkinsonism [169], depression [170,171], dementia [172], Lesh-Nyhan syndrome, Niemann Pick type C, and Multiple Sclerosis [173]. Supporting these findings, [174] found in 2015 also in patients with Allen-Herdon-Dudley syndrome, Prader-Willi syndrome, or asparagine synthetase deficiency lowered levels of HVA and/or 5-HIAA. Generally, low HVA and/or 5-HIAA or high HVA and normal 5-HIAA have been shown in 6–20% of patients suffering from encephalopathies, meningitis, mitochondrial diseases, pontocerebellar hypoplasia, leukodystrophies, de-/hypomyelination, neuropsychiatric conditions, Rett syndrome, Angelman syndrome, and hypoxic ischemic encephalopathy. 

High levels of neopterin in CSF have been reported in different conditions not based on primary neurotransmitter biosynthesis defects. These include early infantile epileptic encephalopathy, Aicardi–Goutières syndrome, HIV infection (also reported with high neopterin and low 5-MTHF), and other immune disorders, as well as inflammation. It has been shown by [175] in 2003 that patients suffering from Aicardi-Goutières Syndrome can present with extremely high levels of neopterin in combination with lowered levels of 5-MTHF. Moreover, in untreated HIV patients, CSF neopterin levels are, in most cases, elevated and increase as immunosuppression becomes worse and the counts for CD4^+^ cells drop [176]. Patients suffering from HIV dementia have particularly high levels of CSF neopterin, which are higher than those measured in infected patients without neurological symptoms. HIV patients with opportunistic infections also show a high level of CSF neopterin. In patients under treatment, intrathecal immunactivation becomes reduced, and CSF neopterin levels are lowered. Nevertheless, neopterin levels in treated patients still remain mildly elevated compared to healthy controls. Neopterin can also be used as marker to distinguish between CNS and peripheral infections in children [177]. The group could show that serum levels of neopterin can be used to identify system infections, but not CNS-localized infections. In contrast, they found high levels of neopterin only in CSF samples of patients suffering from infections present in the CNS, compared to the values of healthy controls. The study also confirmed previous studies showing that peripheral and CNS levels of neopterin are not correlated. Taken together, the authors suggest the use of serum and CNS neopterin levels as an aid in distinguishing between inflammatory versus non-inflammatory disease and between CNS versus peripheral infection. Moreover, neurodegenerative diseases like Parkinson´s disease (PD) are also associated with changes in monoamine neurotransmitter levels. PD is mainly caused by dopamine deficiency in the striatum, which is considered to be due to the loss of nigro-striatal dopaminergic neurons with disease progression. 

Current medication should also be taken into consideration when interpreting the laboratory results initially, but also during treatment monitoring. Drug treatment close to the date of CFS sampling can influence analysis. For example, levodopa (with or without aromatic amino acid decarboxylase inhibitors) can increase l-Dopa, 5-HIAA, and 5-HTOL. Methotrexate can result in a high CSF concentration of phenylalanine combined with low levels of 5-MTHF. Supplementation with folinic acid and sapropterin dihydrochloride (Kuvan^®^) leads to high levels of the related metabolites in the CSF. Also, monoamine oxidase inhibitors and serotonin re-uptake inhibitors can influence biogenic amine levels [87,178,179,180,181]. 

It should be mentioned that pyridoxal phosphate and 5-MTHF should be determined in addition to catecholamines and pterins, in case of suspicion of AADC deficiency and pyridoxine 5’-phosphate oxidase (PNPO) deficiency. In these disorders, high levels of vanillylactic acid in urine can occur besides the classical CSF pattern (low HVA, 5-HIAA, MHPG, elevated 3-OMD) [182]. Generally, patients suffering from AADC deficiency not only suffer from central catecholamine depletion, but also from depletion of 5-HT, because AADC converts both l-Dopa and 5-HTP to dopamine and 5-HT, respectively. As a consequence, in the patients, besides low levels of HVA, 5-HIAA and MHPG, high levels of 5-HTP and 3-OMD can be measured. 3-OMD accumulates if l-Dopa is not converted by AADC but by COMT. Furthermore, monitoring of 5-MTHF should be considered under AADC inhibition and 5-HTP therapy because of secondary folate depletion. 

As a marker of central dopaminergic deficiency, high prolactin levels in plasma can be used, but normal prolactin levels do not exclude a neurotransmitter disorder [183]. It should be taken into consideration that hyperprolactinemia can have other reasons such as physiological or pathological endocrine conditions (hyperhtyroidism, prolactinoma), hypothalamus and pituitary disorders, systemic disorders, infections, drug related changes, renal failure, cirrhosis, and post-ictal status [184,185,186,187].

## 5. Conclusions and Future Perspective

There exists a broad spectrum of analytical tools for the determination of biogenic amines and pterins in clinically relevant specimen, including urine, blood, CSF, and patient-derived fibroblasts. This allows for choosing the best method that applies to a specific project or clinical application. Nevertheless, development of new and fast techniques is ongoing. These include targeted and untargeted metabolomics applications and NMR analyses. However, new techniques have also been developed, including the usage of quantum dots [188,189], Cu (II)-based metal-organic xerogels [190], or gold nanorods [191], for the determination of dopamine in human samples or fluorescence sensor arrays for the quantification of different neurotransmitters [192]. The practicability of these new methods in the context of clinical application has to be determined but holds great promise for a faster and more precise analysis. Moreover, with the dropping costs for next generation sequencing applications, whole genome and whole exome sequencing becomes more and more clinical routine. This allows—and will allow in the future—for the identification of patient-specific yet unknown disease-causing variants, and even new disease entities not directly linked to metabolic pathways but sharing features with metabolic diseases. One example is the recently described DNAJC12 deficiency, which is based on the mutation of a co-chaperone but presenting with hyperphenylalaninemia usually associated with PKU [2,3,4]. In the future, more and more of these “non-metabolic” causes of metabolic disorders are to be discovered. These might also include defects in genes involved in signal transduction or gene regulation (like the proteins involved in DNA methylation and demethylation). Nevertheless, it has to be kept in mind that every variant of unknown significance (VUS) which is identified in a patient has to be thoroughly analyzed on a functional level for its role in disease generation and/or progression. To achieve this, highly specific methods for the determination of enzymatic activity based on the measurement of the amount of the product have to be applied or, if not already present, need to be developed based on already available methods. For “non-metabolic” causes (e.g., not based on variants in an enzyme or transporter), as is the case for DNAJC12 deficiency, new routes to monitor protein activity/functionality have to be established. 

Also, other high throughput analytical methods, including metabolomics (targeted and untargeted), will play an important role for diagnostic and therapeutic purposes in the future, and allow for a more precise surveillance of the applied therapies. This will provide new applications for personalized medicine, starting from the identification of an individual VUS, going via precise metabolic profiling in the patient, and leading to a personalized therapy and therapy monitoring based on the genetic and metabolic data of the patient, with the help of highly sensitive new analytical methods like quantum dots, nanorods, xerogels [190,191,192], newly developed ELISA assays, but also ultrafast HPLC methods like the pterinomics workflow described by Burton et al. in 2016 or the single-step protocol for the rapid analysis of catecholamines, pterins, and serotonin in one sample [40]. 

## Figures and Tables

**Figure 1 cells-08-00867-f001:**
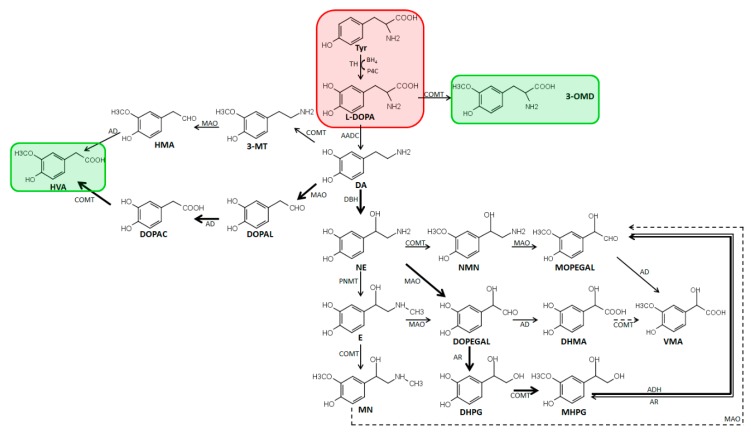
Catecholamine biosynthesis and metabolism. AADC, aromatic amino acid decarboxylase; AD, aldehyde dehydrogenase; ADH, alcohol dehydrogenase; AR, aldehyde reductase; COMT, catechol-*O*-methyltransferase; DA; dopamine; DBH, dopamine ß-hydroxylase; DHPG, 3,4-dihydroxyphenylglycol; DHMA, 3,4-dihydroxymandelic acid; DOPAC, 3,4-dihydroxyphenylacetic acid; DOPAL, 3,4-dihydroxyphenylacetaldehyde; DOPEGAL, 3,4-dihydroxyphenylglycolaldehyde; E, epinephrine; 5HIAA, 5-hydroxyindolacetic acid; HMA, 4-hydroxy-3-methoxyphenylacetaldehyde; HVA, homovanillic acid; L-DOPA, L-3,4-dihydroxyphenylalanine; MAO, monoamine oxidase; MHPG, 3-methoxy-4-hydroxyphenylglycol; 3-*O*-MD, 3-*O*-methyl-DOPA; MN, metanephrine; 3-MT, 3-methoxytyramine; MOPEGAL, 3-methoxy-4-hydroxyphenylglycolaldehyde; NE, norepinephrine; NMN, normetanephrine; PNMT, phenolethanolamine-*N*-methyltransferase; TH, tyrosine hydroxylase; Tyr, tyrosine; VMA, vanillylmandelic acid. Marked in red is the BH_4_ consuming reaction forming L-DOPA. Boxed in green are the most important analytes used for diagnosis in cerebrospinal fluid (CSF). Adapted from [11].

**Figure 2 cells-08-00867-f002:**
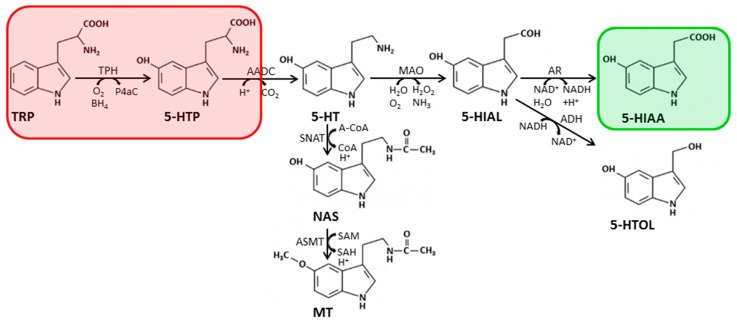
Biosynthesis and catabolism of tryptophan. A-CoA, acetyl coenzyme A; AADC, aromatic amino acid decarboxylase; ADH, alcohol dehydrogenase; AR, aldehyde reductase; ASMT, *N*-acetyl-serotonin-*O*-methyltransferase; CO_2_, carbon dioxide; CoA, coenzyme A; 5-HIAA, 5-hydroxyindolacetic acid; 5-HIAL, 5-hydroxyindolacetaldehyde; 5-HT, 5-hydroxytryptamine (serotonin); 5-HTOL, 5-hydroxytryptophol; 5-HTP, 5-hydroxytryptophan; H^+^, proton; H_2_O, water; H_2_O_2_, hydrogen peroxide; MAO, monoamine oxidase; MT, melatonin; NAD, nicotinamide dinucleotide; NAS, *N*-acetyl-serotonin; NH_3_, ammonia; O_2_, oxygen; P4aC, pterin-4-carbinolamine; SAM, *S*-adenosyl-methionine; SAH, *S*-adenosyl homocysteine; SNAT, serotonin-*N*-acetyltransferase; TRP, tryptophan. Marked in red is the BH_4_ consuming reaction forming 5-HTP. Boxed in green is 5-HIAA, the most important analyte used for diagnosis in CSF.

**Figure 3 cells-08-00867-f003:**
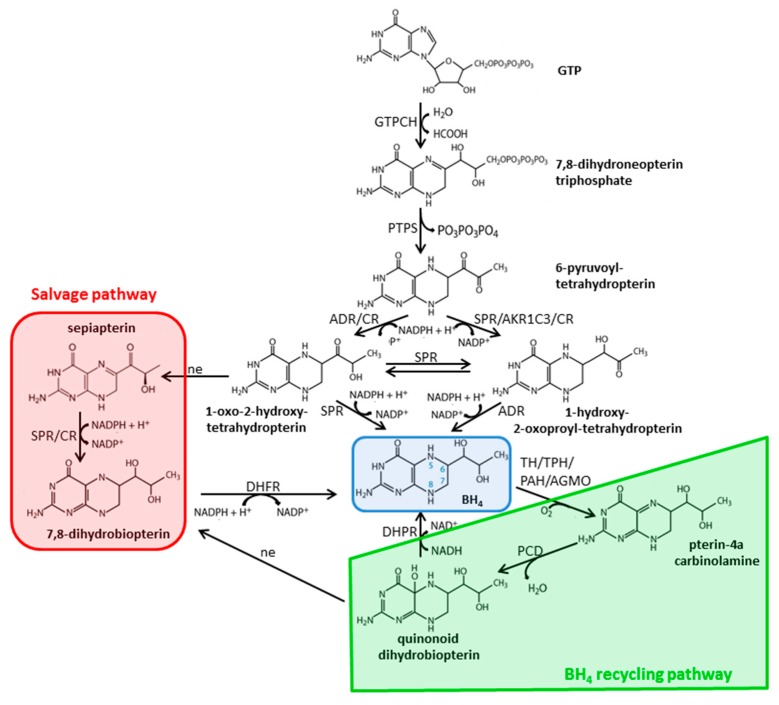
Biosynthesis and recycling of BH_4_. ADR, aldose reductase; AGMO, alkylglycerolether monooxygenase; AKR1C3, 3α-hydroxysteroid dehydrogenase type 2; BH_4_, 5,6,7,8 tetrahydrobiopterin; CR, carbonyl reductase; DHFR, dihydrofolate reductase; DHPR, dihydropteridine reductase; GTP, guanosine triphosphate; GTPCH, GTP cyclohydrolase 1; H_2_O, water; NAD, nicotinamide dinucleotide; NADPH, nicotinamide dinucleotide phosphate; ne, non-enzymatic; O_2_, oxygen; PAH, phenylalanine hydroxylase; PCD, pterin-4-carbinolamine dehydratase; PTPS, 6-pyruvoyl tetrahydrobiopterin synthase; SPR, sepiapterin reductase; TH, tyrosine hydroxylase; TPH, tryptophan hydroxylase.

**Table 1 cells-08-00867-t001:** Sample pre-treatment for the quantification of catecholamines and some of their metabolites in different human fluids.

Matrix	Volume (mL)	Sample Preparation	Solvent	Analytes	Ref.
**Human CSF**	1.0	Dilution	6 mM L-cysteine/2 mM oxalic acid/1.3% glacial acetic acid	DA; DOPAC; HVA; HT; HIAA	[37]
	6th–8th mL	Dilution	0.03% formic acid	L-Dopa; 3-MT; HVA; HIAA; MHPG; 5-HTP	[38]
	2nd tube (400th–800th µL)	Dilution; filtration	NA	HVA; 5-HIAA; 3-OMD; MHPG; 5-HTP	[39]
	50 µL	Dilution; filtration	250 nM 2,5-dihdroxybenzoic acid; 5000 MWCO PES Vivaspin 500 filter	HVA; 5-HIAA; 3-OMD; 5-HTP; MHPG	[40]
**Human plasma**	0.5	SPE	Water/acetonitrile (40:60, *v*/*v*) with 2.5% formic acid	NE; E	[27]
	0.5	PP, TFC	10% TCA; A: Water/0.1% perfluoroheptanoic acid; B: Water/acetonitrile (40:60, *v*/*v*); C: Isopropanol/acetone/acetonitrile (1:1:1) with 0.3% formic acid; D: Water/5 mM ammonium acetate/50% acetonitrile	NMN/MN	[41]
	0.2	SPE	Acetonitrile with 2% formic acid	NMN; MN	[42]
	0.5	SPE	Water/acetonitrile (40:60, *v*/*v*)	NMN; MN	[43]
	0.1	SPE	Water/acetonitrile (5:95, *v*/*v*) with 2% formic acid	NMN; MN; 3-MT	[44]
	0.015	Dilution	10 mM glutathione/10 mM citric acid/100 mgL^−1^ EDTA pH4.5	DA; NE; E; NMN; MN; 3-MT	[45]
	0.5	PP/filtration	1.2 M perchloric acid	L-Dopa; DA; DOPAC	[46]
	0.5	SPE	Aqueous solution (10.5 g L^−1^ citric acid/20 mg L^−1^ EDTA)/acetonitrile (98:2, *v*/*v*); pH 2.8, 1 M NaOH	L-Dopa; DA; NE; E; DHPG	[47]
	0.02	SPE	0.6 M potassium chloride/acetonitrile (2:3, *v*/*v*)	DA; NE; E	[48]
	0.5	SPE	10.5 g L^−1^ citric acid/20 mg L^−1^ OSA/20 mg L^−1^ EDTA/methanol (97.5:2.5, *v*/*v*) pH 2.9, 1 M NaOH	DA; NE; E;	[24]
	0.5	SPE	10.5 g L^−1^ citric acid/20 mg L^−1^ OSA/20 mg L^−1^ EDTA/methanol (95:5, *v*/*v*) pH 3.5, 1 M NaOH; MHPG: methanol	DA; NE/E; MHPG	[29]
	0.1/0.5	LLE	Ammonia buffer/heptane mixture; 80 mM acetic acid/octanol; MHPG: Ethyl acetate	DA; NE; E; MHPG (free and conjugated)	[49]
**Human urine**	0.25	SPE	Water/methanol (95:5, *v*/*v*) with 2% formic acid	NMN; MN	[50]
	3.0	SPE	Water/acetonitrile (20:80, *v*/*v*) with 1% formic acid	DA; NE; E	[51]
	0.04	Filtration	NR	L-Dopa; DA; NE; E, MN	[52]
	20	PBA affinity column	0.1 M phosphate buffer/1 mM EDTA/300 mg L-1 SOS/ methanol (10:1, *v*/*v*), pH 2.5	DA/NE/E	[53]
	0.02	SPE	50 mM potassium dihydrogenphosphate/2.5 mM OSA/0.1 g L-1 EDTA/acetonitrile (96.5:3.5, *v*/*v*); pH 3.5, phosphoric acid	DA; NE; E	[25]
	5.0	SPE	6 M acetic acid	DA/NE/E	[54]
	0.3	LLE	Ammonia buffer/heptane mixture; 166 mM aqueous acetic acid/1-octanol	DA/NE/E	[55]
	1.0	SPE	1 M acetic acid	DA; NE; E	[30]
	5.0	Bio-Rex 70 resin	4 M formic acid	DA; NE; E; NMN; MN	[56]
	5.0	Bio-Rex 70 resin	4 M formic acid	DA; NE; E; NMN; MN	[57]
	0.5	LLE	Ammonia buffer/heptane mixture; 80 mM acetic acid/1-octanol; MHPG: Ethyl acetate	DA; NE; E; MHPG (free and conjugated)	[49]
**Amniotic fluid**	0.2	Dilution	20 mM phosphate buffer; pH 3.0; 0.5 mM heptasulfonic acid; 0.12 mM EDTA; 0.28% perchloric acid; 15% methanol	HIAA; HVA	[58]

DA, dopamine; DOPAC, 3,4-dihydroxyphenylacetic acid; E, epinephrine; EDTA, ethylenediaminetetraacetic acid; 5-HIAA, 5-hydroxyindoleacetic acid; 5-HT, 5-hydroxytryptamine; 5-HTP, 5-hydroxytryptophane; HVA, homovanillic acid; L-DOPA, L-3,4-dihydroxyphenylalanine; LLE, liquid–liquid extraction; MHPG, 3-methoxy-4-hydroxyphenylglycol; MN, metanephrine; NE, norepinephrine; NMN, normetanephrine; NR, not reported; OSA, 1-octanesulfonic acid sodium salt monohydrate; PBA, phenylboronic acid; PP, protein precipitation; SDS, sodium dodecyl sulphate; SOS, sodium octylsulphate; SPE, solid-phase extraction.

**Table 2 cells-08-00867-t002:** Liquid chromatography techniques for the quantitative measurement of catecholamines and some of their metabolites in different human fluids.

Matrix	Technique	Analytes	Sample Preparation	Internal Standard	Column	Mobile Phase	Elution	Detection	Ref.
**Human CSF**	HPLC	DA; DOPAC; HVA; 5-HT; 5-HIAA	Dilution	NR	ESA MD-150 C18	75 mM monobasic sodium phosphate buffer/0.5 mM EDTA/0.81 mM OSA/5% tetrahydrofuran/acetonitrile (95:5, *v*/*v*) pH 3.1; phosphoric acid	Isocratic	Coulometric	[37]
	HPLC	DA; NE; DOPAC; HVA; MHPG; 5-HT; 5-HIAA	Direct injection dialysate	NR	Luna C18	0.2 M phosphate buffer pH 5.0	Isocratic	Amperometric	[84]
	HPLC	3-OMD; HVA; 5-HIAA; MHPG; 5-HTP	Dilution, filtration	3-OMD, HVA; 5-HIAA; MHPG; 5-HTP	ODS (C18)	0.1 M sodium acetate; 0.1 M citric acid; 1.2 mmol/l EDTA, 1.2 mmol/l 1-heptanosulfonic acid; 75 mL methanol; adjusted to pH 4.0	Isocratic	Coulometric	[39]
	UHPLC	HVA; 5-HIAA; 3-OMD; 5-HTP; MHPG	Direct injection	HVA; 5-HIAA; MHPG; 3-OMD; 5-HTP	ACQUITY UPLC HSS T3	0.05 M citrate buffer; pH 5.2; methanol (97:3, *v*/*v*)	Isocratic	Coulometric; FL ex: 350 nm; em: 450 nm	[40]
**Human plasma**	LC	NE; E	SPE	d6-NE; d6-E	C18	A: 10 mM ammonium formate in water; B: Methanol	Gradient	MS/MS; [positive ionization electrospray]	[27]
	LC	NMN; MN	PP; TFC	d3-NMN; d3-MN	Hypercarb PGC	A: 50 mM ammonium formate/1% formic acid in water; B: 0.1% formic acid in acetonitrile; C: Isopropanol/acetone/acetonitrile (9:2:9, *v*/*v*/*v*); D: 0.1% perfluoroheptanoic acid in water	Gradient	MS/MS; [positive ionization electrospray]	[41]
	UHPLC	NMN; MN	SPE	d3-NMN; d3-MN	Atlantis HILIC	A: Acetonitrile; B: 200 mM ammonium formate pH 3.0	Gradient	MS/MS; [positive ionization electrospray]	[42]
	LC	NMN; MN	SPE	d3-NMN; d3-MN	Hypercarb PGC Hypersil Gold HILIC	A: 50 mM ammonium formate/1% formic acid in water; B: 0.1% formic acid in acetonitrile; C: Isopropanol/acetone/acetonitrile (9:2:9, *v*/*v*/*v*) A: 100 mM ammonium formate/acetonitrile (5:95, *v*/*v*) pH 3.2; B: Acetonitrile/water/100 mM ammonium formate (50:45:5, *v*/*v*/*v*) pH 3.2	Gradient	MS/MS; [positive ionization electrospray]	[43]
	HPLC	NMN; MN; 3-MT	SPE	d3-NMN; d3-MN; d4-3-MT	Atlantis HILIC	A: 100 mM ammonium formate in water pH 3.0; formic acid; B: Acetonitrile	Gradient	MS/MS; [positive ionization electrospray]	[44]
	HPLC	DA; NE; E; NMN; MN; 3-MT	Dilution	MHBA	Unison UK-C18	75 mM potassium acetate buffer/100 mM potassium phosphate buffer/8 mM sodium 1-hexanesulfonate/acetonitrile (93.1:4.9:2, *v*/*v*/*v*); pH 3.2	Isocratic	CL; [TDPO/H_2_O_2_]	[45]
	HPLC	L-DOPA; DA; DOPAC; 3-O-MD; Carbidopa	PP/Filtration	DHBA	ESA HR-80 C18	Modified CAT-A-PHASE buffer: Phosphate buffer/patented ion-pairing agent/methanol/acetonitrile (99.7:0.3, *v*/*v*); pH 3.2; 2 N NaOH	Isocratic	Coulometric	[46]
	HPLC	L-DOPA; DA; NE; E; DHPG	SPE	DHBA	Deverosil RPAQUEOUS-AR-5 C30	Aqueous solution of 10.5 g L^−1^ citric acid/20 mg L^−1^ EDTA/acetonitrile (98:2, *v*/*v*); pH 2.8; 1 M NaOH	Isocratic	Amperometric	[47]
	HPLC	DA; NE; E; MHPG	SPE	DHBA	Microsorb C8	10.5 g L^−1^ citric acid/20 mg L^−1^ EDTA/20 mg L^−1^ OSA/methanol (95:5, *v*/*v*); pH 3.5; 1 M NaOH	Isocratic	Amperometric	[29]
	HPLC	DA; NE; E	SPE	DHBA	Rainin C8	25 mM citric acid/20 mg L^−1^ EDTA/20 mg L^−1^ OSA/methanol (97:3, *v*/*v*) pH 2.9; 1 M NaOH	Isocratic	Amperometric	[28]
	HPLC	DA; NE; E	SPE	IPT	TSK gel ODS-120 T	120 mM imidazole buffer/methanol/acetonitrile (13:4:18, *v*/*v*/*v*) pH 5.8	Isocratic	CL	[48]
	HPLC	DA; NE; E	SPE	DHBA	Jones Apex C8	10.5 g L^−1^ citric acid/20 mg L^−1^ EDTA/20 mg L^−1^ OSA/methanol (97.5:2.5, *v*/*v*); pH 2.9; 1 M NaOH	Isocratic	Coulometric	[24]
	HPLC	NE; E	LLE	DHBA	Hypersil ODS	9.02 g sodium acetate/0.372 g EDTA/100 mg SDS/methanol (85:15) or (80:20) pH 5.1; glacial acetic acid	Isocratic	Amperometric	[85]
**Human urine**	LC	NMN; MN	SPE	d3-NMN d3-MN	Ultra II PFP propyl	0.2% formic acid/methanol (95:5, *v*/*v*)	Isocratic	MS/MS; [positive ionization electrospray]	[50]
	HPLC	L-DOPA; DA; NE; E; MN; 5-HT; tryptophan; andderivative	Filtration	NR	Fluofix-II 120E	Water/acetonitrile/trifluoroacetic acid (40:60:0.05, *v*/*v*/*v*)	Isocratic	FL; [PFOEI]; ex: 280 nm; em: 320 nm	[52]
	HPLC	L-DOPA; DA; NE; E; DOPAC	Dilution	IPT	Kromasil C18	A: Methanol; B: 0.1 M sodium acetate buffer pH 5.0; acetic acid	Gradient	FL; [DPE]; ex: 350 nm; em: 480 nm	[86]
	HPLC	DA; NE; E	SPE	NR	ZIC-HILIC; BEH-amide	6.5 mM ammonium formate/acetonitrile (25:75, *v*/*v*) pH 3.0; 6.5 mM ammonium formate/acetonitrile (15:85, *v*/*v*) pH 3.0	Isocratic	Coulometric	[51]
	LC	NMN; MN	SPE	d3-NMN d3-MN	Atlantis T3 C18	A: 10 mM ammonium formate/1% formic acid; B: Methanol	Gradient	MS/MS; [positive ionization electrospray]	[59]
	HPLC	DA; NE; E	SPE	d4-DA; d3-NE; d3-E	Allure PFP propyl	A: 25 mM ammonium formate in water pH 3.0; formic acid; B: Methanol	Gradient	MS/MS; [positive ionization electrospray]	[26]
	HPLC	DA; NE; E	PBA affinity column	NR	Nucleosil C18	0.1 M phosphate buffer/1 mM EDTA/300 mg L^−1^ SOS/methanol (10:1, *v*/*v*) pH 2.5	Isocratic	Amperometric	[53]
	HPLC	DA; NE; E	SPE	NR	Lichrosorb LC-8 C8	50 mM potassium dihydrogen phosphate/500 mg L^−1^ SDS/250 mg L^−1^ EDTA/100 mL L^−1^ methanol/ 200 mL L^−1^ acetonitrile pH 3.5; orthophosphoric acid	Isocratic	CL; [luminol–I2]	[60]
	HPLC	DA; NE; E	SPE	DHBA	RECIPE reversed-phase	50 mM potassium dihydrogen phosphate/2.5 mM OSA/0.1 g L^−1^ EDTA/acetonitrile (96.5:3.5, *v*/*v*) pH 3.5; phosphoric acid	Isocratic	Amperometric	[25]
	HPLC	DA; NE; E	SPE	DHBA	Hypersil-BDS	50 mM acetate buffer/0.11 mM EDTA/1.1 mM OSA/methanol (85:15, *v*/*v*) pH 4.7; 8.5 M acetic acid	Isocratic	FL [TbCl_3_]; ex: 300 nm; em: 545 nm	[54]
	HPLC	DA; NE; E	LLE	d4-DA; d3-NE; d3-E	Allure Basix	6.5 mM aqueous formic acid/tetrahydrofuran (2:3, *v*/*v*)	Isocratic	MS/MS; [positive ionization electrospray]	[55]
	HPLC	DA; NE; E	SPE	DHBA	Luna C18	50 mM dihydrogen phosphate buffer/500 mg L^−1^ SDS/250 mg L^−1^ EDTA/100 mL L^−1^ methanol/200 mL L^−1^ acetonitrile pH 2.9; 6 M orthophosphoric acid	Isocratic	Coulometric	[30]
	HPLC	DA; NE; E; NMN; MN	LLE	DHBA	Spherisorb C8	A: Acetonitrile; B: 3.0 g L^−1^ acetic acid solution	Gradient	FL; [FMOC-Cl]; ex: 263 nm; em: 313 nm	[74]
	HPLC	DA; NE; E; NMN; MN	SPE	DHBA	Nova-Pak C18	50 mM ammonium formate; pH 3.0; formic acid	Isocratic	APcI-MS	[56]
	HPLC	DA; NE; E; NMN; MN	Cation exchange resin	DHBA	Nova-Pak C18	200 mM NaH_2_PO_4_·H_2_O/0.2 g L^−1^ EDTA/4 mM sodium 1-heptanesulfonate/acetonitrile (97.8:2.2, *v*/*v*) pH 3.0; 1 M orthophosphoric acid	Isocratic	Amperometric	[57]
	HPLC	NE; E; 5-HIs	Dilution/filtration	5-HIA	Cosmosil 5C18	10 mM acetate buffer/acetonitrile (65:35, *v*/*v*) pH 6.0	Isocratic	FL; [benzylamine]; ex: 345 nm; em: 480 nm	[80]
**Amniotic fluid**	HPLC	5-HIAA; HVA	NA	NA	Hypersil 3MOS	20 mM phosphate buffer; pH 3.0; 0.5 mM heptasulfonic acid; 0.12 mM EDTA; 0.28% perchloric acid; 15% methanol	Isocratic	Amperometric	[58]

APcI-MS, atmospheric pressure chemical ionization mass spectrometry; CL, chemiluminescence; CSF, cerebrospinal fluid; DA, dopamine; DHBA, 3,4-dihydroxybenzylamine; E, epinephrine; EDTA, ethylenediaminetetraacetic acid; FL, fluorescence; 5-HIA, 5-hydroxyindole-3-acetamide; 5-HIs, 5-hydroxyl indoles; 5-HIAA, 5-hydroxyindoleacetic acid; 5-HTP, 5-hydryxytryptophane; HPLC, high performance liquid chromatography; 5-HT, 5-hydroxytryptamine; HVA, homovanillic acid; IPT, isoproterenol; IS, internal standard; LC, liquid chromatography; l-DOPA, 3,4-dihydroxyphenylalanine; LLE, liquid–liquid extraction; 3-OMD, 3-*O*-methyl-DOPA; MHBA, 3-methoxy-4-hydroxybenzylamine; MN, metanephrine; MS/MS, tandem mass spectrometry; 3-MT, 3-methoxytyramine; NE, norepinephrine; *N*-MeDA, *N*-methyldopamine; NMN, normetanephrine; NR, not reported; OSA, 1-octanesulfonic acid sodium salt; SPE, solid-phase extraction; TDPO, bis[2-(3,6,9-trioxadecanyloxycarbonyl)-4-nitrophenyl]oxalate.

**Table 3 cells-08-00867-t003:** Typical CSF profiles of HVA, 5-HIAA, and 3-OMD in inborn errors of biogenic amine metabolism (before treatment initiation).

Deficiency	HVA	5-HIAA	3-OMD
**Tyrosine hydroxylase**	low	normal	normal
**AADC**	low	low	high
**DBH**	high	normal	normal
**Pterin deficiency (recessive)**	low	low	normal
**Pterin deficiency (dominant)**	normal to low	normal to low	normal
**DAT **	high	normal	NR
**VMAT2**	normal	normal	NR

3-OMD, 3-*O*-methyl DOPA; AADC, aromatic amino acid decarboxylase; DAT; dopamine transporter; DBH, dopamine ß hydroxylase; 5-HIAA, 5 hydroxyindolacetic acid; HVA, homovanillic acid; NR, not reported; VMAT2, vesicular monoamine transporter 2.

**Table 4 cells-08-00867-t004:** Sample pre-treatment for the measurement of pterins in different human fluids.

Matrix	Volume (mL)	Sample Preparation	Solvent	Analytes	Ref.
**Human CSF**	1 mL	Precipitation; oxidation	33 mg TCA/1 mg DTT per mL CSF; 0.1 mL HCl (0.1 M)/0.2% iodione/0.4% potassium iodide; 1% ascorbic acid; 1 M HCl/1 mg MnO_2_ per 200 µL CSF	N; B	[38]
	30 µL	Stabilization; oxidation	DTT; DETAPAC	BH_4_; BH_2_; N; S	[108]
	100 µL	Filtration	NA	BH_4_; BH_2_; DHN; B; N	[118]
	3rd tube (800th–1200th µL)	Dilution; filtration	NA	BH_4_; BH_2_; N	[39]
	50 µL	Dilution; filtration	250 nM 2,5-dihdroxybenzoic acid; 5000 MWCO PES Vivaspin 500 filter	BH_4_; BH_2_; B; N; DHN	[40]
**Human plasma**	400 µL	Precipitation; oxidation	1 M TCA; 0.5% iodine/1% potassium iodide/0.2 M TCA, 1% ascorbic acid (biopterin); 6; sodium hydroxide/0.5% iodine/1% potassium iodide/0.2 M TCA; 1% ascorbic acid/6 M sodium hydroxide (BH_4_)	B; BH_4_	[107]
	100 µL	Protein precipitation; derivatization; liquid phase extraction; drying; reconstitution	Ice-cold acetonitrile; 500 mM ammonium carbonate; benzoyl chloride; ethyl acetate; hexane; acetonitrile	BH_4_	[119]
	4 mL	Precipitation; oxidation; purification	2N TCA/0.5% iodione/1% potassium iodide in 0.2 N TCA; Dowex 50 column	B; N	[110]
**Human serum**	200 µL	Oxidation; deproteinization	1 M HCl with 1 mg MnO_2_; Ultrafree (NMWL 10000)	B; N	[120]
	2 mL	Oxidation; ion exchange	I_2_ (5 g/L) in 0.2 M TCA or I_2_ (5 g/L) and KI (10 g/L) in 0.1 M NaOH; AGMP-50 (200–400 mesh (H^+^))	B; N	[121]
	3 mL	Filtration; oxidation; ion exchange; evaporation	0.22 µM nylon mesh; 3 M TCA/2% iodione/4% potassium iodide; ISOLUTEENV; elution in acetonitrile/water (80/29, *v*/*v*); dissolved in mobile phase	PCA; X; N; M; ISO; P; 6-B; 7-B; 6-HMP	[122]
**Dried blood spots**	4 blood spots	Extraction; sonication; ultrafiltration	250 µL 20 mM HCl; Ultrafree (NMWL 10000)	N; B; ISO; P	[123]
	2 blood spots	Extraction; sonication; ultrafiltration	250 µL 20 mM HCl; Ultrafree Nanosep 10 Ω	B; N	[124]
**Erythrocytes**	NA	Washing; lysis; deproteinization; oxidation	154 mM NaCl; water; 1.84 M TCA; 1 M HCl with 1 mg MnO_2_	B; N	[120]
**Human urine**	500 µL	Oxidation; filtration	6 M HCl/10 mg MnO_2_	P; ISO; 6-B; 7-B; 6-N; 7-N	[125]
	100 µL	Oxidation; filtration	4% potassium iodide/2% iodine solution (w/v)	6-B; 6-HMP; N; P; ISO; X	[126]
	400 µL	Oxidation; filtration	A: 2 M NaOH, iodide/iodine solution; B: 5 mM KMnO_4_	B; N; P; PCA; 6,7 DMP; ISO; X; 6-HMP	[127]
	100 µL	Filtration	A: Lugol’s solution (4% iodide/2% iodine solution (w/v)), B: MnO_2_; C: Potassium permanganate	P; X; 7,8-DX; ISO; 6-B; S; N; M; 6-CP; 6-HMP; 6,7-DMP; 6-MP; 6-HLU; 7-HLU; 6-FP; L	[128]
	100 µL	Acidification of urine; oxidation	0.5% iodine/1% iodide in alkaline and acidic solution	B; N; P; BH_4_; BH_2_	[129]
	360 µL	Stabilization; filtration	1% ascorbic acid; Nanosep 10Ω	S	[103]
	500 µL	Dilution	Citrate buffer 10 mM; pH 5.5	N; B; P; ISO	[113]
	1 mL	Acidification; oxidation; extraction	6N HCl; iodide/iodine solution (in 0.1 N NaOH or 0.1 N HCl); Dowex 50W X8, elution with 0.5 M NH4OH; Dowex 1 X8, elution in 1 N acetic acid	X; N; B; BH_4_	[130]
	500 µL	Acidification; oxidation	6 M hydrochloric acid; MnO_2_ (10 mg);	B; N; M	[131]
	1 µL (injection volume)	Dilution; SPE	20× in 1% DTT	BH_4_; BH_2_; N; DHN	[132]
**Amniotic fluid**	200 µL	Acidic oxidation; deproteinization	MnO_2_; 30% TCA	N; M; ISO; B; PR; P	[58]
		Oxidation; clean-up with ion exchange resin	Iodide/iodine at pH 1.0	B; N	[133]
	200 µL	Oxidation; precipitation	1 M hydrochloric acid; 2 m MnO_2_; 30% TCA	B; N	[134]
**Cell lysates**	80 µL	Lysis and sonication	0.2 M TCA, 50 mM ascorbic acid; 1 mM EDTA; 6.5 mM DTE	BH_4_; BH_2_; B	[117]
	150 µL	Lysis; oxidation; deproteinization	50 mM Tris-HCl; pH 7.5; 1 mM DTT (lysis); acidic iodine (10 g/L)	B; N	[115]
	350 µl per pellet	Lysis; sonication; protein isolation	50 mM potassium phosphate buffer; pH 7.0; 0.2 mM PMSF; fast desalting column	N	[116]
	NA	Lysis; sonication; deproteinization; oxidation	Extraction buffer (20 mM Tris-HCl, pH 7.4, 0.1 mM EDTA, 1 mM DTT, 10% glycerol, 0.1% Tween 20); 30% TCA; MnO_2_ (10 mg) with 0.2 M H_3_PO_4_	B; N	[114]

B, biopterin; 6-B, 6-biopterin; 7-B, 7-biopterin; BH_4_, 5,6,7,8, tetrahydrobiopterin; BH_2_, 7,8 dihydrobiopterin; DHN, dihydroneopterin; 6-CP, 6-carboxpterin; 6,7-DMP, 6,7-dimehtylpterin; DTT, dithiothreitol; 7,8 DX, 7,8-dihydroxanthopterin; 6-FP, 6-formylpterin; 6-HMP, 6-hydroxymethylpterin; 6-HLU, 6-hydroxylumazine; 7-HLU, 7-hydroxylumazine; ISO, isoxanthopterin; L, leucopterin; LU, lumazine; 6-MP, 6-methylpterin; M, monapterin; NA, not addressed; N, neopterin; 6-N, 6-neopterin; 7-N, 7-neopterin; PCA, pterin 6-carboxylic acid; P, pterin; PMSF, phenylmethanesulfonyl fluoride; PR, primapterin; S, sepiapterin; TCA, trichloroacetic acid; X, xanthopterin.

**Table 5 cells-08-00867-t005:** Liquid chromatography techniques for the quantitative measurement of pterins in different human fluids.

Matrix	Technique	Analytes	Sample Preparation	Internal Standard	Column	Mobile Phase	Elution	Detection	Ref.
**Human CSF**	HPLC	B; N	Dilution	NA	UltraPure Torsic Acid	4.6 g/L NH_4_HPO_4_; pH 3.5	Isocratic	FL; ex: 350 nm; em: 450 nm	[38]
	HPLC	N; BH_4_; BH_2_; S	Dilution	^15^N-BH_4_; ^15^N-BH_2_; ^15^N-N	AAA-MS column	A: 0.1% formic acid/0.1% heptafluorobutyric acid in water; B: 0.1% formic acid in methanol	Gradient	MS/MS; [positive ionization electrospray]	[108]
	HPLC	BH_4_; BH_2_; DHN; B; N	Dilution	LU	Atlantis dC18	0.005 M sodium citrate/methanol (97:3; *v*/*v*)	Isocratic	Post-column oxidation; FL: ex: 350 nm; em: 450 nm	[118]
	HPLC	BH_4_; BH_2_; N	Dilution, filtration	BH_4_; BH_2_; N	ODS (C18)	50 mM sodium acetate; 5 mM citric acid; 48 µM EDTA; 160 µM DTE	Isocratic	FL; ex: 360 nm; em: 440 nm (oxidized pterins); coulometric (reduced pterins)	[39]
	UHPLC	BH_4_; BH_2_; B; N; DHN	Direct injection	BH_2_; B; N; DHN	ACQUITY UPLC HSS T3	0.05 M citrate buffer; pH 5.2 (pH 7.4 for BH_4_); methanol (97:3, *v*/*v*)	Isocratic	Coulometric, FL ex: 350 nm; em: 450 nm	[40]
**Human plasma**	HPLC	B; BH_4_	Dilution	NA	Hypersil C18	A: 15 mM potassium phosphate buffer; pH 6.45; B: Methanol	Gradient	FL; ex: 360 nm; em: 440 nm	[107]
	HPLC	BH_4_	Derivatization	BH_4_-benzoyl chloride-d5	HILIC polar imidazole column	A: Acetonitrile/water (15%/85%; *v*/*v*) with 0.2% formic acid; B: Acetonitrile with 0.2% formic acid	Gradient	MS/MS; [positive ionization electrospray]	[119]
	HPLC	B; N	Dilution	N	Whatman 10 µm ODS; Partisil 10; µBondapak C18	5% or 20% methanol in water	Isocratic	FL; ex: 350 nm; em: 410 nm	[110]
**Human serum**	HPLC	B; N	Cation exchange resin	BH_4_; BH_2_	Excalibur ODS	Methanol/water (15/85, *v*/*v*)	Isocratic	FL; ex: 370 nm; em: 418–700 nm	[121]
	HPLC	PCA; X; N; M; ISO; P; 6-B; 7-B; 6-HMP	Cation exchange	PCA; X; N; M; ISO; P; 6-B; 7-B; 6-HMP	Zorbax-Eclipse XDB C18 and Poroshell 120	2 mM ammonium formiate; pH 7.1	Isocratic	FL; ex: 272 nm; em: 410 nm, 445 nm, 465 nm	[122]
**Dried blood spots**	HPLC	B; N; ISO; P	None	B; N; P	Pre column C8 Spherisorb; ODS-1 Spherisorb	1.5mM potassium hydrogen phosphate buffer; pH 4.6/8% methanol (*v*/*v*)	Isocratic	FL; ex: 350 nm; em: 450 nm	[123]
	UPLC	B; N	Dilution	^13^C_5_-N; D3-B	ACQUITY UPLC HSS T3	A: 0.2% formic acid in water; B: 0.2% formic acid in methanol	Gradient	MS/MS; [positive ionization electrospray]	[124]
**Human urine**	HPLC	P; ISO; 6-B; 7-B; 6-N; 7-N	Dilution	BH_4_; P; ISO; 6,7-DMP	ZORBAX C18; LUNA amino; HILIC; AQUA C18	Methanol/0.1% formic acid; acetonitrile/0.1% formic acid; water/0.1% formic acid/10 mM ammonium formiate	Isocratic	MS/MS; [negative ionization electrospray]	[125]
	HPLC	6-B; 6-HMP; N; P; ISO; X	Dilution	6-B; 6-HMP; N; P; ISO; X	Phenyl-hexyl column	A: 0.1% formic acid in H_2_O; B: 0.2% formic acid in acetonitrile	Gradient	MS/MS; [positive ionization electrospray]	[126]
	HPLC	B; N; P; PCA; 6,7 DMP; ISO; X; 6-HMP	Dilution	B; N; P; PCA; 6,7 DMP; ISO; X; 6-HMP	LiChrospher C8 60 RP; Aquasil C18; HILIC Luna	A: 10 mM phosphate buffer, pH 7.0; B: Methanol (LiCrospher); A: 10 mM Tris-HCl, pH 6.8; B: Methanol (Aquasil); A: 100 mM ammonium acetate buffer pH 5.8; B: Acetonitrile (HILIC)	Gradient (LiCrospher; HILIC); isocratic (Aquasil)	FL; ex: 280 nm; em: 444 nm; UV: 215 nm and 254 nm	[127]
	HPLC	P; X; 7,8-DX; ISO; 6-B; S; N; M; 6-CP; 6-HMP; 6,7-DMP; 6-MP, L; 6-HLU; 7-HLU; 6-FP; L	Dilution	P; X; 7,8-DX; ISO; 6-B; S; N; M; 6-CP; 6-HMP; 6,7-DMP; 6-MP; L; 6-HLU; 7-HLU; 6-FP; L	Luna	A: 0.025% (*v*/*v*) formic acid in 99% water/1% acetonitrile; B: Methanol	Gradient	MS/MS; [positive ionization electrospray]	[128]
	HPLC	B, N, P, BH_4_, BH_2_		6-MP	RP 18	Water/methanol (97/3, *v*/*v*)	Gradient	FL; ex: 350 nm; em: 410 nm	[129]
	HPLC	N; B; P; ISO	Post-column photo derivatization	BH_4_; BH_2_	Pre-column: XDB-C18; Zorbax Eclipse XDB-C18	Citrate buffer; pH 5.5/acetonitrile (98/2, *v*/*v*)	Isocratic	FL; ex: 272 nm; em: 445 nm; photometric: 256 nm	[113]
	HPLC	S		X	Spherisorb S5 ODS1250	A: 24 mM K_2_HPO_4_, pH 5.0; B: Methanol/water (70/30; *v*/*v*)	Gradient	FL; ex: 425 nm; em: 530 nm	[103]
	HPLC	X; N; B; BH_4_	None	6-MP	Partisil-10 SCX	1 mM ammonium dihydrogen phosphate; pH 2.8/7% methanol/5% acetonitrile	Isocratic	FL; ex: 360 nm; em: 420 nm	[130]
	HPLC	B; N; M	Dilution	B; M; N; P	Spherisorb S5 ODS	A: Methanol/water (3:97, *v*/*v*); B: Isopropanol/methanol/acetic acid (49:49:2, *v*/*v*/*v*); C: Isopropanol-methanol-water (1:1:8, *v*/*v*/*v*); D: 6.6 mM Na_2_HPO_4_/13.3 mM citric acid/0.06 mM Na_2_EDTA/1.4 mM octanesulphonic acid/10% methanol, pH 3.3; E: 6.6 mM Na_2_HPO_4_/13.3 mM citric acid/0.06 mM Na_2_EDTA/1.4 mM octanesulphonic acid/10% isopropanol, pH 3.3; F: 20 mM KH_2_PO_4_/0.85 mM octanesulphonic acid/0.1 mM Na_2_EDTA/1% methanol, pH 3.0	Gradient (A, B, C for oxidized pterins); Isocratic (E, F for reduced pterins)	FL; ex: 350 nm; em: 450 nm (oxidized pterins); amperometric (reduced pterins)	[131]
	UHPLC	BH_4_; BH_2_; N; DHN	SPE	BH_4_; BH_2_; N; DHN	BEH Amide column	50 mM ammonium acetate; pH 6.8; acetonitrile (15:85, *v*/*v*)	Isocratic	FL; ex: 353 nm; em: 438 nm; UV detection (PDA detector) at 253 nm	[132]
**Amniotic fluid**	HPLC	B; N	Ion exchange resin	6-MP	Partisil-10 SCX	1 mM ammonium dihydrogen phosphate; pH 2.8/7% methanol/5% acetonitrile	Isocratic	FL; ex: 360 nm; em: 420 nm	[133]
**Cell lysates**	HPLC	BH_4_; BH_2_; B	None	^15^N-BH_4_; ^15^N-BH_2_; ^15^N-B	Poroshell 120 SB-C18 column	A: 150 mM acetic acid, 12 mM HFBA; B: 12 mM HFBA, methanol	Gradient	MS/MS; [positive ionization electrospray]	[117]
	HPLC	B; N	Deproteinization	B	Spherisorb C18	ND	Gradient	FL; ex: 350 nm; em: 450 nm	[115]
	HPLC	N	GTPCH activity assay	N	Inartsil ODS-3	10 mM sodium phosphate; pH 7.0/1 mM EDTA	Isocratic	FL; ex: 365 nm; em: 475 nm	[116]
	HPLC	B; N	NA	NA	Econosphere C18	0.1 M sodium phosphate; pH 3.0/5% methanol	Isocratic	FL; ex: 350 nm; em: 450 nm	[114]

B, biopterin; 6-B, 6-biopterin; 7-B, 7-biopterin; BH_4_, 5,6,7,8, tetrahydrobiopterin; BH_2_, 7,8 dihydrobiopterin; DHS, deoxysepiapterin; DHN, dihydroneopterin; 6-CP, 6-carboxpterin; 6,7-DMP, 6,7-dimethylpterin; 7,8 DX, 7,8-dihydroxanthopterin; 6-FP, 6-formylpterin; 6-FDP, 6-formyldihydropterin; 6-HMP, 6-hydroxymethylpterin; 6-HLU, 6-hydroxylumazine; 7-HLU, 7-hydroxylumazine; ISO, isoxanthopterin; L, leucopterin; LU, lumazine; 6-MP, 6-methylpterin; M, monapterin; ND, not described; N, neopterin; 6-N, 6-neopterin; 7-N, 7-neopterin; PCA, pterin 6-carboxylic acid; PR, primapterin; P, pterin; S, sepiapterin; X, xanthopterin.

**Table 6 cells-08-00867-t006:** Pterin profiles in different body fluids in BH_4_ deficiencies.

Matrix	Analytes	arGTPCHD	adGTPCHD	PTPSD	DHPRD	SPRD	PCDD
**Human urine**	biopterin	low	low to normal	low	normal (to high)	normal	low to normal
	neopterin	low	low to normal	high	normal (to high)	normal	normal to high
	xanthopterin	NR	NR	NR	(high)	NR	NR
	primapterin	NR	NR	NR	normal	normal	high
	sepiapterin	NR	NR	NR	NR	high	NR
**Human CSF**	biopterin	low	low	low	normal (to high)	normal (to high)	normal
	BH_4_	low	low to normal	low	low to normal	low	NR
	neopterin	low	low	high	normal	normal	normal
	BH_2_	(low)	NR	NR	normal (to high)	high	NR
	primapterin	(normal)	NR	NR	NR	NR	NR
	sepiapterin	(normal)	NR	NR	NR	high	NR
**Dried blood spot**	biopterin	low	low to normal	low	normal to high	normal	NR
	neopterin	low	low to normal	high	normal to high	normal	NR
	primapterin	NR	NR	NR	normal	normal	NR

ad/arGTPCHD, autosomal dominant/autosomal recessive GTP cyclohydrolase deficiency; DHPRD, dihydropteridine reductase deficiency; NR, not reported; PCDD, pterin-4 -carbinolamine dehydratase deficiency; PTPSD, 6-pyruvoyl tetrahydrobiopterin synthase deficiency; SRD, sepiapterin reductase deficiency; () few reported cases.

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
