# Peer review of "Analysis of Catecholamines and Pterins in Inborn Errors of Monoamine Neurotransmitter Metabolism—From Past to Future"

_cells, 2019, doi:10.3390/cells8080867_

Round 1

Reviewer 1 Report

This manuscript will be a useful addition to the area of disorders of monoamine and pterin metabolism.  The title suggests discussion of future approaches.  Discussion around the future currently makes up a small proportion of the manuscript.    The authors should be encouraged to expand this section, e.g. personalized  medicine, impact of genetic testing, need for bespoke functional testing when variants of unknown significance are detected.  Speculation  around new disorders to be discovered could be included.

1. The authors should cross reference with the relatively  recent publication of Batllori et al., 2017. 

2. 5-hydroxytryptophan should be mentioned.  Also how this is elevated along with 3-methyldopa in AADC deficiency.

3. A clear section on pitfalls should be included.  This could also include drug effects - Monoamine oxidase inhibitors, Re-uptake inhibitors, methotrexate, L-dopa and 5-hydroxytryptophan.  Also secondary contributor factors such as hypoxia -ischemia and neurodegerneration.

4.Direct quantification of BH4 in CSF is described as challenging.  A detailed methods is described and if, like the other methods considered, a protocol is adhered to then it is entirely possible quantitate BH4.  Again the authors should reference Batllori et al for the protocol.

5. The authors should make it clear when assessment of Pyridoxal phosphate and methyltetrahydrofolate may be required.  Also the usefulness of Vanillactate in urine and prolactin measurements.

6. Table A1 should include dopamine transporter defects.

Reviewer 2 Report

I read with great interest the the manuscript entitled " Analysis of catecholamines and pterins in inborn 2 errors of monoamine neurotransmitter metabolism – 3 from past to future".

The aim of this report is to summarize number of commonly used analytical tools of diagnosis  for quantitative determination of neurotransmitters and cofactors in the different types of samples used to identify patients affected by certain rare diseases.

 Here are some comments to the authors : 

1-  limitations and strengths.

a- The paper is informative and didactic.
b- The tables contain relevant, practical and well-documented informations.

c- It would be beneficial to mention, whenever they exist, the limitations of the different tools mentioned.

2- It's worth pointing out the fact that the approach of dosing sepiapterin in the urine allows the diagnosis of SR deficiency in a peripheral sample therefore avoiding the lumbar puncture (Carducci et al. Mol Genet Metab 2015).

3- Pterins are determined primarily by chromatographic techniques with fluorescence or electrochemical detection (Blau and al. (2008), Laboratory guide to the methods in biochemical genetics, Springer). While this approach allows the differentiation of the different genetic conditions related to BH4 deficiency; authors could mention alternative methods, even if they are not used routinely, such as capillary electrophoresis coupled to fluorescence detection, an online UV-photoirradiation HPLC system, employing a a diode array and fast scanning fluorescence detectors and ultra-PLC with fluorescence detection (Tomšíková et al. J Pharm Biomed Anal 2014; Cañada-Cañada, Anal Chim Acta 2009).

4- in paragpah 3.1.4 Analysis of catecholamines in CSF : HVA and 5-HIAA can be used as indirect markers to explore dopamine and serotonin pathways dysfunction in the brain. Authors should complete this section :

The analyses of 3-ortomethyldopa and methoxyhyroxyphenylglycol (as dopamine and norepinephrine metabolites) and 5-hydroxytryptophan, as a serotonin precursor metabolite are proving to be quite valuable enabling  identification and differentiation of the biogenic amine disorders in a single analysis.

5- In paragraph 3.1.5. : Analysis of monoamines in urine should not be used to diagnose genetic disorders because 5-HIAA or HVA can vary considerably depending on external factors.

6- In paragraph 3.1.6  authors indicate that blood samples should be collected into a tube containing an antioxidant and an anticoagulant, transported on ice, centrifuged at 4°C and plasma stored at -80°C until measurement.
which anticoagulant and which antioxidant can they recommend ?

7- In paragraph 3.2.5 authors indicate that the native sample should be stored at -20 ° C and protected from light until processed for analysis.
According to some other sources, the urine sample should be protected from light to prevent degradation, and suggest oxidizing before storing the sample at –80ºC until analysis (Blau and al. (2008), Laboratory guide to the methods in biochemical genetics, Springer), what is the best temperature range at which the samples are to be stored ?

8- The authors may mention the works of Lo and al. (ACS Omega. 2017 )  describing a method that allows determing in a single step  neurotransmitter metabolites as well as pterins of interest by UHPLC coupled to sequential electrochemical and fluorescence detections.

9-  5-HIAA ou la 5-HT contained in the table  have not been subject to an accurate synthesis (synthesis and degradation), such as catecholamines or pterins. Some lines would be appropriate  on synthesis and degradation of 5-HT.

10- While diagnosis of Inborn errors of monoamine neurotransmitter biosynthesis and degradation is predominantly based on quantitative detection of neurotransmitters, cofactors and precursors in a sample of biological fluid, one can only regret there has been no paragraph with an indication of inborn errors of metabolism mimicking monoamines dysfunction. Indeed, several studies have shown the presence of secondary alterations in biogenic amines in the CSF of neurological patients ( 6–20% of the studied patients) (García-Cazorla an al., Dev Med Child Neurol 2007; Molero-Luis and al. Dev Med Child Neurol 2013; Van Der Heyden and al. Eur J Paediatr Neurol 2003; Mercimek-Mahmutoglu and al. Orphanet J Rare Dis 2015). Some patients with neurometabolic diseases might exhibit low levels of HVA and 5-HIAA in CSF, even in concentrations lower than primary defects, meaning that an overlap can exist in the biogenic amine values of primary and secondary defects. Indeed, The relevant of clinical, radiological, and, obviously, genetic information must not be overlooked (Molero-Luis an,d al. Dev Med Child Neurol 2013).

Reviewer 3 Report

The paper is detailed and clearly written. However, it is quite dense with information about methods and would be improved if the 'Profile' tables in the appendix were in the main article. The main purpose of running the methods is for diagnosing and monitoring metabolic disorders so I think the profile tables should be given more prominence.

Tables A1 and A2 contain a number of errors and omissions. These include:-

-  spelling (pterines - please be consistent)

- dopamine transporter and VMAT defects should be added to A1

- BH4 is usually normal-low in CSF in adGTPCHD and low in SPRD and PTPSD (Table A2)

- BH2 is usually high in CSF in DHPRD and SPRD (Table A2)

- Neopterin should also be included in table A2 and it is useful for diagnosing GTP cyclohydrolase deficiencies (often low) and PTPS deficiency (elevated).

If tables A1 and A2 are corrected and given more prominence, the article would be improved.

I would also suggest that other conditions affecting monoamine metabolism should be discussed (e.g. Parkinson's disease etc.) and how these methods are also useful for treatment monitoring as well as diagnosis. How is a profile affected if the patient was on L-dopa or sapropterin/Kuvan treatment.

Example chromatograms would also be a useful addition.
